# Local negative feedback of Rac activity at the leading edge underlies a pilot pseudopod-like program for amoeboid cell guidance

Jason P. Town[1,2], Orion D. Weiner[1,2]*

1 Cardiovascular Research Institute, University of California, San Francisco, San Francisco, California, United States of America, 2 Department of Biochemistry and Biophysics, University of California, San Francisco, San Francisco, California, United States of America

* orion.weiner@ucsf.edu

## Abstract

To migrate efficiently, neutrophils must polarize their cytoskeletal regulators along a single axis of motion. This polarization process is thought to be mediated through local positive feedback that amplifies leading edge signals and global negative feedback that enables sites of positive feedback to compete for dominance. Though this two-component model efficiently establishes cell polarity, it has potential limitations, including a tendency to "lock" onto a particular direction, limiting the ability of cells to reorient. We use spatially defined optogenetic control of a leading edge organizer (PI3K) to probe how neutrophil-like HL-60 cells balance "decisiveness" needed to polarize in a single direction with the flexibility needed to respond to new cues. Underlying this balancing act is a local Rac inhibition process that destabilizes the leading edge to promote exploration. We show that this local inhibition enables cells to process input signal dynamics, linking front stability and orientation to local temporal increases in input signals.

## Introduction

Neutrophils find and kill invading pathogens by dynamically aligning their front-back polarity axis with subtle external chemoattractant gradients that indicate injury or infection. One of the fundamental problems these cells need to solve is the consolidation of the biochemical signals that drive protrusions to a small portion of the cell surface. To navigate their complex environments efficiently, migrating cells must balance this "decisive" consolidation process with the "flexibility" needed to continuously update the direction of this polarity axis.

Cells consolidate their protrusive activity through a reaction–diffusion Turing system [1] that combines local positive and global negative feedback [2–6]. In neutrophils, the positive feedback loop is linked to protrusion signaling and involves phospholipids (including phosphatidylinositol 3,4,5-triphosphate (PIP$_3$)), small GTPases (including Rac), and f-actin [7–9]. Cell protrusions push on the plasma membrane, generating increases in tension that rapidly equilibrate throughout the cell [10] and stimulate global inhibition of Rac activation and actin polymerization [5,11]. This reaction–diffusion model explains several key cell behaviors,

located at the Zenodo database (https://doi.org/10.5281/zenodo.8208724). • The analysis notebooks are at the Zenodo database (https://zenodo.org/record/8217762). • The microscopy automation code is located at the Zenodo database (https://zenodo.org/record/8217768).

**Funding:** This work was supported by the National Institute of General Medical Sciences (GM118167 to ODW), the National Science Foundation (2019598 and DBI-1548297 Center for Cellular Construction to ODW and a Predoctoral Fellowship to JPT), and a Novo Nordisk Fonden grant (NNF17OC0028176 Center for Geometrically Engineered Cellular Systems to ODW). The funders had no role in study design, data collection and analysis, decision to publish, or preparation of the manuscript.

**Competing interests:** The authors have declared that no competing interests exist.

**Abbreviations:** iLID, improved light-induced dimer; PIP$_3$, phosphatidylinositol 3,4,5-triphosphate; PI3K, phosphoinositide 3-kinase; TIRF, total internal reflection fluorescence.

including the consolidation of protrusive activity into a single front and the ability to detect chemoattractant gradients over a wide range of ambient concentrations. However, this model has a fundamental limitation: Although simulated cells can properly orient to an initial gradient, they tend to "lock" and have difficulty reorienting their polarity when correcting errors in orientation or when the external gradient changes [3].

As a potential solution to the locking tendency of local-positive-feedback-global-negative-feedback systems, Meinhardt proposed the existence of an additional factor, a local inhibitor, that slowly destabilizes fronts of migrating cells, preventing "front-locking" [3]. Modeling approaches have made use of this sort of local inhibition to produce realistic simulated chemotactic behaviors, such as pseudopod-splitting in *Dictyostelium* [12]. Modeling can be used to show how these signaling topologies could function even without knowledge of the specific molecular components. Furthermore, with suitable inputs, the action of a local inhibitor can be demonstrated experimentally even if the specific molecular candidate is not known [13]. Experimental studies have advanced our understanding of several potential molecular candidates of a Meinhardt-style local inhibitor [14–18]. However, it is unclear whether these candidates act locally due to possible confounding effects from other globally acting inhibitors. To formally demonstrate the existence of a local inhibitor, these confounding effects would need to be removed from the system [19]. Perturbations to global inhibition, however, typically disrupt the organization of the front signals that represent the substrates of the local inhibitor [20–22]. One way to demonstrate the existence and signaling logic of front-based local inhibition would be to experimentally control these front signals to "reorganize" them following perturbations to global inhibition.

Here, we investigate how cells decode intracellular guidance cues by leveraging an optogenetic approach to control the production of PIP$_3$, a key protrusion-activating signal in migrating neutrophil-like HL-60 cells [7,8,23,24]. Using this strategy in combination with dynamic, computer-controlled spatial stimulation and pharmacological perturbations, we demonstrate the existence of a local negative feedback loop operating on Rac at the leading edge of migrating neutrophil-like cells. We show that this local negative feedback loop enables these cells to detect the local rate of signal increase, consistent with the pilot pseudopod model for cell guidance, first described by Gerisch [25]. To explain our results, we propose a modified pilot pseudopod model that accounts for insensitivity at the backs of cells [21,23,26–29] and for saturated signals at the fronts of migrating cells. These two modifications are necessary to explain the particular sensitivity of the lateral edges of migrating cells to PIP$_3$ signals. The edges are within the sensitive region and have more "room to grow" compared to the center of the front, possibly due to having less local inhibition. This "peripheral attention" model explains how cells use temporal features of internal signals to continuously steer their fronts and balance decisiveness and flexibility during navigation.

## Results

### Local opto-PI3K stimulation steers migrating HL-60 cells

Previous work has found that local activation of phosphoinositide 3-kinase (PI3K) signaling induces the generation of actin-based protrusions such as pseudopodia and growth cones [10,24,30]. Since polarized cells differ in the biochemical compositions of their fronts and backs, we first sought to understand whether the response to local PI3K activation spatially varied in migrating HL-60 cells. Toward this end, we used an optogenetic approach to control PI3K signaling (opto-PI3K) in time and space. This enabled us to probe migrating cells' biochemical and directional responses to user-defined changes in PIP$_3$ at different subcellular locations (**S1A Fig**).

Our opto-PI3K approach uses an improved light-induced dimer (iLID) system [31] to recruit endogenous PI3K to the plasma membrane of HL-60 cells in response to blue light (Fig 1A). We first verified the effectiveness of our optogenetic system using spatially patterned blue light stimulation (470 nm) and PHAkt-Halo, a live-cell biosensor for $PIP_3$ production [32–34] in stationary cells (S1B–S1D Fig and S1 Video). Upon local activation of $PIP_3$ production (as evidenced by local accumulation of PHAkt-Halo), we observed a concurrent local increase in the localization of Pak-PBD-mCherry. This live-cell biosensor predominantly recognizes active Rac in HL-60 cells [35] and typically localizes to the fronts of migrating cells. Inspired by previous work in smart microscopy control systems [36–40], we implemented a computer vision–based feedback system that tracks the dynamic features of migrating cells (primarily location and movement, biosensor polarity, and visibility in total internal reflection fluorescence (TIRF) microscopy) and delivers automatically updated, spatially patterned optogenetic stimuli (Fig 1B). Using this system, we can reproducibly track and perturb $PIP_3$ signals in time and space in moving cells.

Local opto-PI3K stimulation at a given subcellular location (e.g., side, front, back) caused reproducible behaviors across migrating cells. Compared to unstimulated, freely migrating cells (Fig 1C), local optogenetic stimulation of PI3K signaling at one lateral edge of a migrating cell tended to cause that cell to turn toward and persistently migrate in the direction of the optogenetic stimulus (Fig 1D). Local activation at the fronts of cells (S2A Fig) caused highly directed movement in the direction of stimulation, and this motion was more directed than either unexposed control (Fig 1C) or uniformly stimulated cells (S2B Fig). Local activation at the backs of cells (S2C Fig) caused them to either perform a "u-turn" [41] or depolarize, in which case the cell footprint disappeared from the TIRF plane. The directional stimuli generally caused continuous turning responses as opposed to a depolarization–repolarization response (S2 Video), and the rates of these turning responses varied depending on the location of the stimulus (S2D Fig). We could even induce cells to turn continuously by updating the position of the stimulus to continuously target one edge of the cell (S3 Video). To characterize the biochemical responses of these cells, we collected fluorescence data from the periphery of moving cells and mapped it onto a standard frame of reference (angle from cell centroid) to enable averaging within and comparisons between conditions (Figs 1E, 1F, and S3). With these controllable perturbations and measurable outcomes, we could now ask which spatial and temporal features of $PIP_3$ signaling contributed to these cell behaviors.

## Sensitivity to local $PIP_3$ stimulation is spatially biased to the lateral edges in polarized cells

We next sought to determine whether responsiveness to optogenetic stimulation and downstream activation varied spatially in polarized cells, which are known to differ in the biochemical compositions of their fronts and backs. As other researchers have observed with various signaling inputs, we found the backs of cells to be relatively insensitive to stimulation via opto-PI3K [21,23,26–29]. Cells took longer to reorient (S2D Fig) in the direction of the stimulus and displayed u-turn behaviors in response to stimulation at the back. Restricting signaling competency to the fronts of cells could be one mechanism to ensure persistent movement, though, at an extreme, this strategy could cause polarity locking [3]. To quantify the spatial extent of this bias in cells moving in an approximately two-dimensional environment, we took advantage of our ability to simultaneously stimulate multiple locations in moving cells using our computer vision–driven system. This enabled us to expand upon results from previous work on back insensitivity for cells constrained in one-dimensional channels [28,29].

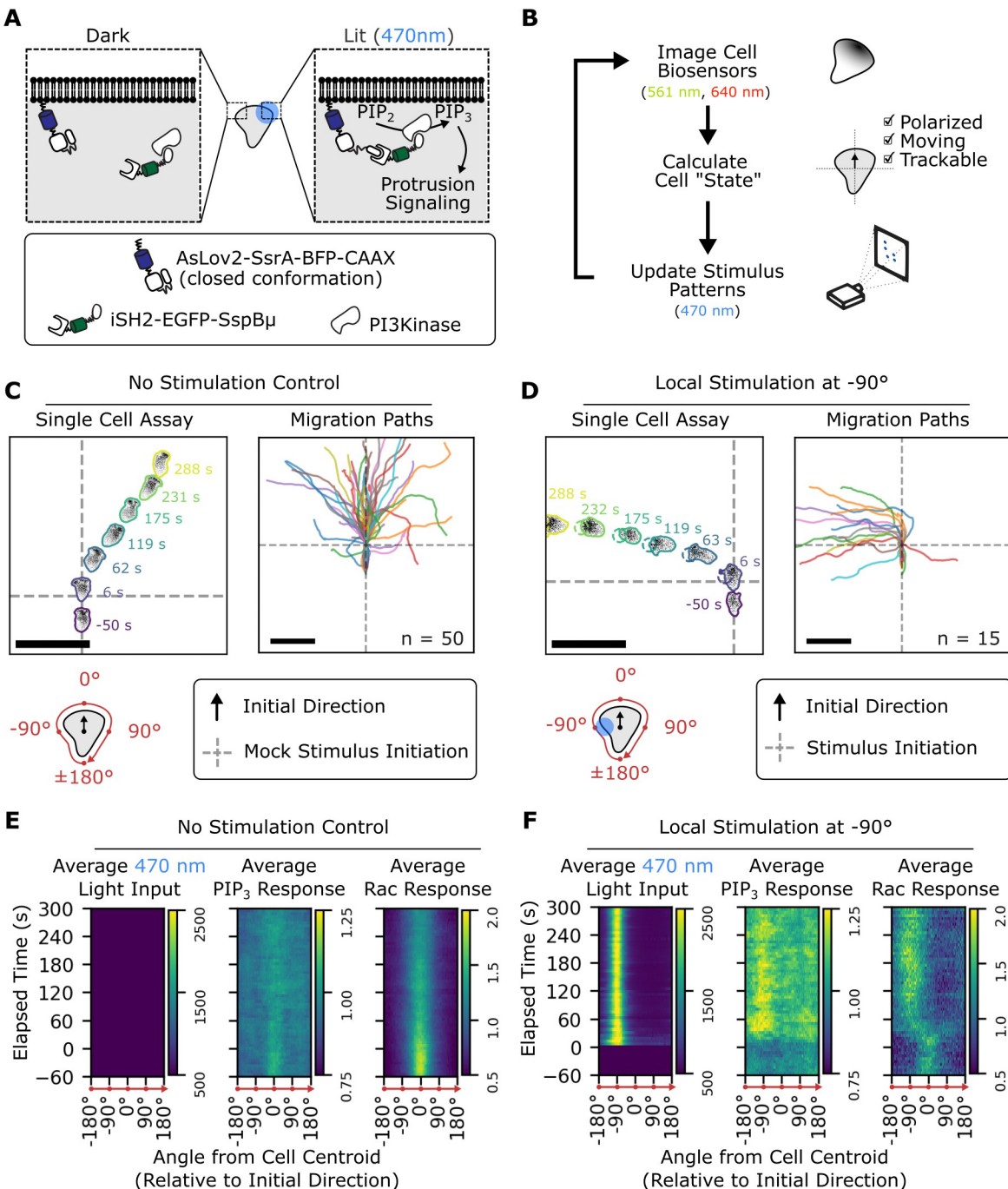

**Fig 1. Computer-guided, spatially controlled optogenetic stimulation of PI3K signaling controls cell directionality. (A)** Schematic depicting spatial control of PI3K recruitment using the iLID optogenetic system. A membrane-bound, light-responsive "anchor" (AsLov2-SsrA-BFP-CAAX) exposes the SsrA peptide upon blue light (470 nm) illumination. This exposed peptide can then recruit SspB. By attaching signaling "cargo" to SspB, localization of the cargo from the cytoplasm to the plasma membrane is placed under blue light control. We used the iSH2 domain of P85 beta (which binds endogenous PI3K) as our signaling cargo. **(B)** Schematic depicting the computer vision–guided, closed-loop feedback control of optogenetic stimulation. This automation enabled us to dynamically reposition optogenetic inputs for moving cells, enabling improved throughput and reproducibility for a given stimulation assay compared to manual inputs. Cells expressed fluorescent translocation-based biosensors for PIP3 (PHAkt-Halo labeled with JF646) and Rac activity (mCherry-Pak-PBD). Since these biosensors were imaged with 640 nm and 561 nm light, respectively, they could be imaged without activating the blue light–sensitive optogenetic system. **(C)** Cells expressing the opto-PI3K constructs (shown in 1A) and biosensors for PIP3 and Rac activity were allowed to migrate freely for 6 minutes. The cells generally maintained their existing directionality during this time. The Rac biosensor is shown in the Single Cell Assay subpanel. In both the Single Cell Assay and the Migration Paths subpanels, the spatial data

have been rotated and translated such that the average directionality in the first minute of imaging is toward the top of the figure, and the location of the cell at the 1-minute mark is at the intersection of the gray, dashed lines. Scale bars: 50 μm **(D)** When cells are stimulated at one side with blue light starting at the 1-minute mark, they reorient their fronts in the direction of the stimulus. The Rac biosensor is shown in the Single Cell Assay subpanel. Scale bars: 50 μm **(E)** In the absence of blue light stimulation, migrating cells maintain their current axis of polarized signaling. These kymographs represent an average of the radial distribution of reflected blue light (Average Light Input, arbitrary units) or normalized fluorescence intensities (Average Normalized PIP$_3$ Response, Average Normalized Rac Response) over time across all measured no-stimulation control cells (as shown in C, upper-right). Peripheries are aligned so that the initial direction of the cell is 0˚. **(F)** In the local stimulation condition, cells reorient both their directionality (as shown in 1D) as well as their biosensor distributions. Like 1E, these kymographs show the averaged radial distribution of measured quantities around the peripheries of migrating cells. There is a local increase in PIP$_3$ at the −90˚ location, corresponding to the site of optogenetic stimulation. The underlying data for this figure can be found in S1 Data. iLID, improved light-induced dimer; iSH2, inter-SH2; PIP$_3$, phosphatidylinositol 3,4,5-triphosphate; PI3K, phosphoinositide 3-kinase.

When we activated both lateral edges of a migrating cell in a 180˚-opposed fashion, cells stably turned toward only one of the two stimuli (**Fig 2A** and **S4 Video**). When the opposed stimuli were oriented perpendicularly relative to the initial direction of motion, cells broke symmetry evenly, with half of the cells migrating to the left and the other half migrating to the right. This effect was not predicted by initial differences in PIP$_3$ accumulation as both the "winning" and "losing" side had similar initial local PIP$_3$ dynamics (**Figs 2B and S4**). By varying the angle of these opposed competing signals relative to the initial axis of movement, we found that cells were more likely to turn toward stimuli closer to the existing front (**Figs 2C and S5**). At the extreme, all cells stimulated simultaneously at their front and back continued moving in their initial direction. These results support the notion that the back of the cell responds less strongly to our optogenetic input.

Cells stably moved toward only one of the two optogenetic stimuli, indicating that this assay created a winner-take-all scenario. Surprisingly, however, the frontward advantage was gradual. In a straightforward winner-take-all system, as in the case of coupled local positive feedback and global negative feedback, any slight initial advantage should be amplified, leading to switch-like advantages. Thus, we might have expected cells to reliably turn toward more front-oriented signals, even for small angles. We sought to resolve this apparent contradiction by further exploring the winner-take-all nature of two-spot competition assays.

## Fronts of polarized cells show behaviors consistent with local negative regulation

The observed stability of migration decisions in response to 180˚-opposed stimuli could represent an intrinsic ability of cells to commit to one site of PIP$_3$ stimulation by suppressing signaling elsewhere. Alternatively, it could represent an initial commitment that becomes stable due to the insensitivity of the back of the cell to stimulation. To differentiate between these possibilities, we altered the stimulation assay to ensure that both stimulation sites would be continuously near the fronts of cells. We accomplished this by comparing cell behaviors for assays in which activation spots were placed at the cell edges either ±90˚ or ±45˚ from the initial axis of movement (**Fig 2D**). In the ±45˚ assay, if cells were to direct their fronts toward one stimulus, the other stimulus would then be located at the lateral edge of the cell, which should be sensitive to stimulation compared to the back of the cell. During the 5 minutes of frontward-oriented opposed-stimulus exposure, 48% of cells in the ±45˚ assay showed some degree of reversal between the two orientations, compared to 8% in the ±90˚ assay (see **S6 Fig** for a description of our classification strategy). This result suggests that the insensitivity of the backs of cells drives the previously observed stability in the ±90˚ spot competition assay. The long-term stability is not due to an intrinsic ability to suppress signaling at a distance. On the contrary, the orientation switching observed in the ±45˚ assay suggests the existence of a factor

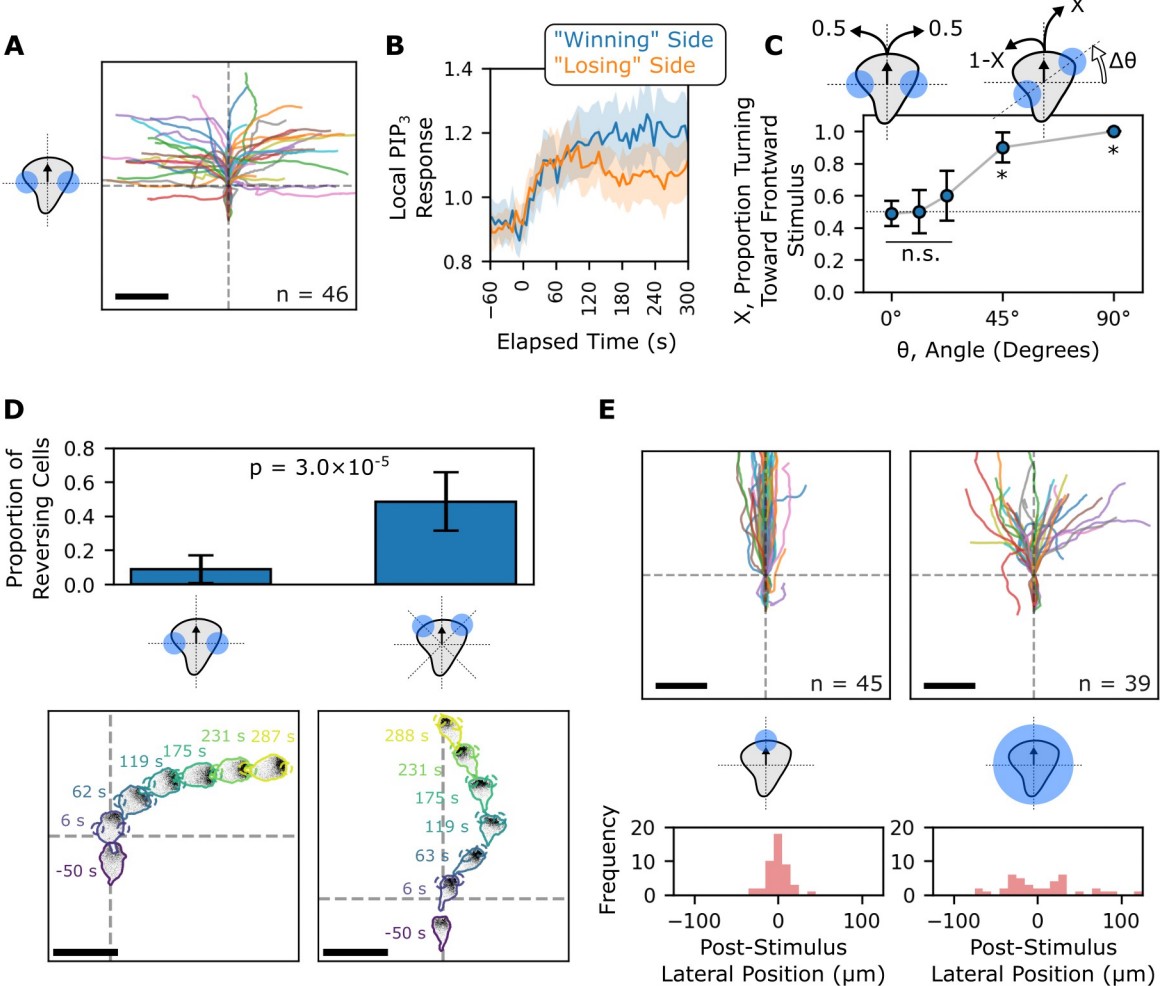

**Fig 2. Optogenetic control of PI3K signaling reveals spatial differences in responsiveness to input signals in migrating cells. (A)** When migrating cells are stimulated simultaneously at both lateral sides (at ± 90˚) with 180˚-opposed blue light opto-PI3K stimuli, they orient themselves in the direction of only one of the two stimuli. Scale bar: 50 μm. **(B)** Local PIP₃ biosensor dynamics are initially similar whether a cell ends up orienting toward a particular stimulation site ("Winning" Side) or away from it ("Losing" Side). Mean local responses are shown with 95% CI shaded. **(C)** When the angle of 180˚-opposed stimulation sites is varied relative to the initial direction of the cell, a preference for frontward signals is revealed. (from left to right, $n = 41$, $n = 14$, $n = 10$, $n = 10$, $n = 6$). n.s.: not significant meaning $p > 0.05$ under the null assumption that cells are as likely to respond to one stimulus as the other. $P < 0.05$ inidicated by * symbol. Proportions shown with 95% CI error bars. **(D)** The stable winner-take-all nature of the 180˚-opposed stimuli is a consequence of spatial effects rather than an intrinsic ability of the cells to ignore a second local stimulus. If two spots are located at ± 45˚ (right) rather than ± 90˚ (left), cells frequently change their spot prioritization during the course of the 5-minute assay. Proportions turning toward frontward stimuli with 95% CI are shown. The *p*-value shown is the result of a two-sample Z-test for proportions. Scale bars: 50 μm. **(E)** The frontward bias shown in 2C does not cause globally stimulated cells (right) to persist in their current direction of movement compared to front-simulated cells (left). Scale bars: 50 μm. The underlying data for this figure can be found in **S1 Data**. PIP₃, phosphatidylinositol 3,4,5-triphosphate; PI3K, phosphoinositide 3-kinase.

that locally inhibits front signaling, thereby destabilizing fronts on a slow timescale and making them less able to suppress signaling at distant sites over time.

One additional piece of evidence in support of a local inhibition at cell fronts is the difference in directional responses of the cells to local front stimulation compared to global stimulation assays (**Fig 2E**). In the global stimulation assay, cells tend to deviate from their initial direction significantly more than cells driven to migrate forward via local optogenetic stimulation. Local inhibition at the center of the fronts of cells could account for these larger responses

at regions just outside this area. This is also apparent when examining fold changes in biosensor enrichment around the peripheries of these cells. In both front-stimulated and globally stimulated cells, Rac signal does not significantly increase at the exact fronts of cells during the first minute of stimulation, but the edges of the fronts do increase (S3C and S3D Fig). Overall, our data are consistent with a combination of insensitivity at the backs of cells [21,23,26–29] and local inhibition–based insensitivity at the extreme fronts of cells.

Our data suggest the existence of local inhibition at the fronts of cells, as predicted by Meinhardt [3]. This front-based inhibition is distinct from previously observed inhibition (or lack of responsiveness) at the backs of cells and is instead expected to enable directional plasticity. To directly demonstrate local negative regulation at the fronts of cells, we next sought to remove confounding effects from global negative feedback.

## Direct demonstration of local negative feedback for Rac activation

In HL-60 cells with an intact cytoskeleton, protrusion generation causes rapid, actin-dependent increases in membrane tension throughout the cell [10]; thus, we expect cells to limit protrusion growth via mechanically mediated global negative feedback [5,11,42,43]. Since both global and local inhibition act on and are acted on by components of the positive feedback loop (Fig 3A), they are difficult to disambiguate in the intact system. To isolate the effects of local inhibition, we blocked actin-based protrusions with latrunculin, preventing activation of mechanically gated global inhibition. Inhibiting this particular node is also expected to interrupt front-based positive feedback [8], leaving only the putative influence of local negative regulation (Fig 3B).

To avoid previously observed oscillatory dynamics of Rac activity in latrunculin-treated differentiated HL-60 cells [20–22], we elected to use undifferentiated HL-60 cells, which retain the ability to generate protrusions [10] in response to opto-PI3K stimulation but are basally quiescent. When we locally stimulated one side of these latrunculin-treated opto-PI3K cell with blue light, we observed a sustained increase in $PIP_3$ (as evidenced by PHAkt enrichment) and a transient increase in Rac activity (as evidenced by Pak-PBD enrichment) (Fig 3C, blue curves in left plots). These dynamic responses were largely confined to the stimulated side of the cell; the side of the cell away from the stimulus experienced much weaker responses of a similar character (Fig 3C, blue curves in right plots). To demonstrate that global negative feedback (long-range inhibition) is not present in this experimental context, we tested whether including second spot of activation showed any evidence of "action at a distance." We found that the local dynamics at the first side were not measurably impacted by the presence of a second stimulus on the opposite side of the cell (Fig 3C, orange curves in left plots). Finally, the global inhibitor is thought to be involved in protrusion-site competition, enabling a winner-take-all behavior. We compared the local responses of migrating cells stimulated with two sites of optogenetic activation with that of latrunculin-treated cells (S7 Fig). In migrating cells, the Rac response shows a mutually exclusive distribution, suggesting a winner-take-all system. In the latrunculin-treated cells, Rac responses are highly correlated, suggesting little-to-no "action-at-a-distance."

These results, combined with the transient nature of the Rac response in latrunculin-treated cells, suggest a nonglobal, actin-independent mode of inhibition. To determine the spatial range of this inhibition, we subjected cells to a local optogenetic stimulus until inhibition was observed and then applied a secondary global stimulus after a brief rest period. During this experiment, we measured the asymmetry in signaling by comparing the dynamics on the pre-stimulated half of the cell and the other side. As expected, local optogenetic recruitment of PI3K resulted in steady and reversible polarization of our $PIP_3$ reporter (Fig 3D, orange

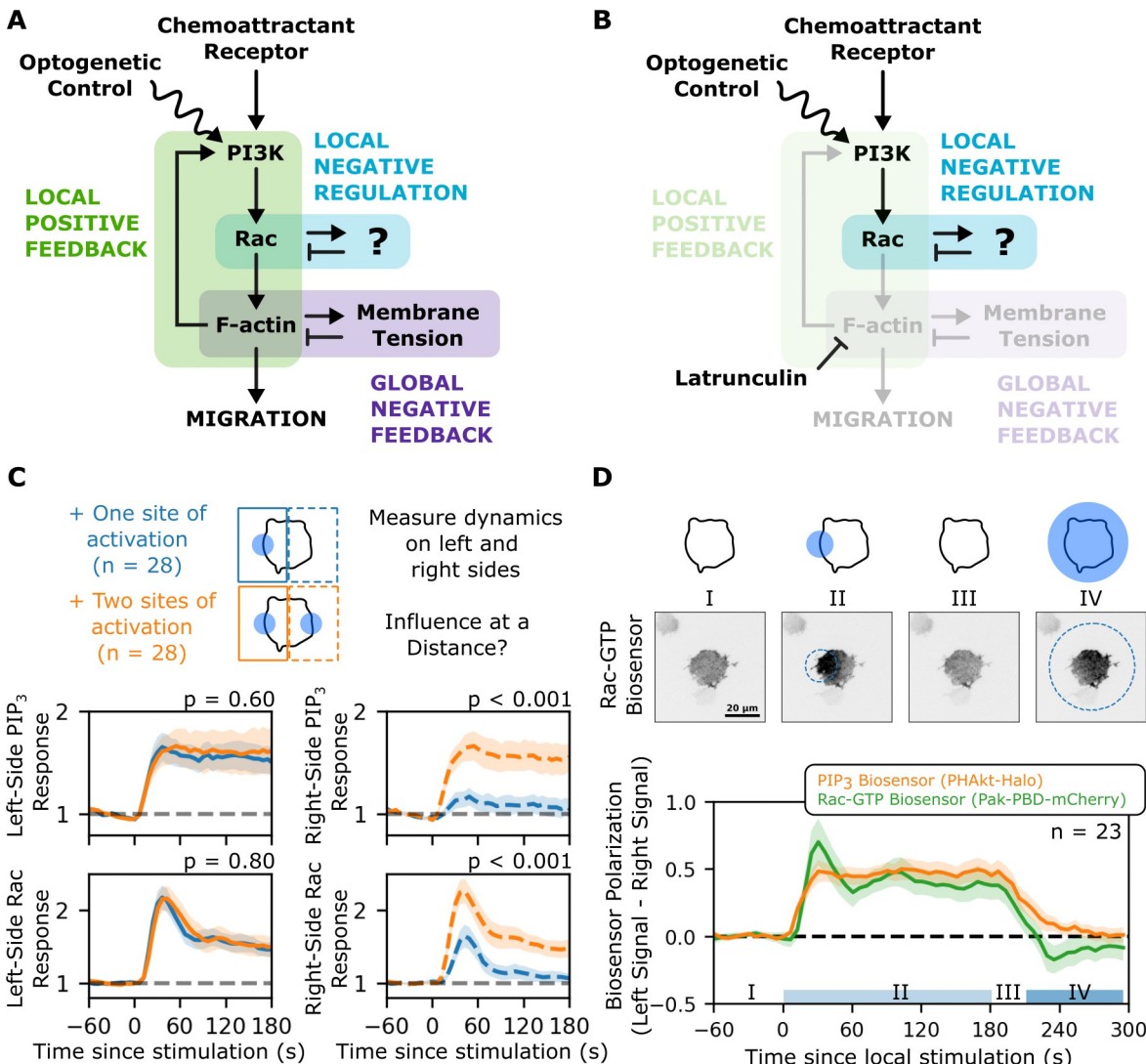

**Fig 3. Using opto-PI3K to reveal locally acting inhibition of Rac activation. (A)** Cell polarity depends on a combination of positive and negative feedback loops with different spatial ranges of action. Locally acting positive feedback (green shaded portion, mediated through phospholipids, GTPases, and actin polymerization) locally amplifies the signals that drive cell protrusion. Actin-based protrusions increase the tension in the plasma membrane to globally inhibit secondary sites of protrusion (purple-shaded region). These two feedback loops (short-range positive, long-range negative) are thought to form the core of a polarity circuit in many cellular contexts but cannot explain the flexibility of cell polarity unless other feedback circuits are also included. An additional locally inhibitory feedback loop has been proposed (blue-shaded region) to avoid the tendency of this core circuit to lock on a given direction of polarity and ignore new stimuli. Direct evidence of this local inhibitor in neutrophil polarity is lacking. **(B)** Because both local positive feedback and global negative feedback are thought to operate through actin-dependent processes, we hypothesized that treating cells with latrunculin B to block actin polymerization would enable us to observe evidence of local inhibition without the confounding effects of these feedback loops. **(C)** To demonstrate the absence of global inhibition in latrunculin-treated cells, we exposed them to either one or two spots of opto-PI3K activation and compared the resulting dynamics on both sides. In the absence of a global inhibitor, the responses for a single opto-PI3K stimulus should be the same whether or not there is a secondary site of opto-PI3K activation (in contrast to competition between these sites in the presence of the global inhibitor, as in Fig 2A–2C). When the right sides of cells are compared in each type of stimulation, the presence or absence of the local right stimulus makes a large difference in the observed signaling dynamics. However, the dynamics on the left sides of cells do not differ based on the presence or absence of a distant site of stimulation. Thus, there is no inhibitory action-at-distance, which would be expected from a global inhibitor. The *p*-values above each graph indicate the probability of observing an absolute difference greater than or equal to that observed by a permutation test. Mean local responses are shown with 95% CI shaded. **(D)** To demonstrate the presence of local inhibition, we first exposed single cells to a local opto-PI3K stimulus on their left sides, then allowed cells to recover briefly, and then exposed them to a global opto-PI3K stimulus. We reasoned that if the decline in signaling on the left side were due to a local inhibitor, the impact of that inhibition should be observable via a weaker response on the prestimulated side. We observed this evidence of local inhibition at the level of Rac activation (green: Rac is biased toward the right side in phase IV) but not at the level of the opto-PI3K input (orange: no bias of PIP₃ to the right side in phase IV). Mean local responses are shown with 95% CI

shaded. The underlying data for this figure can be found in **S1 Data**. PIP$_3$, phosphatidylinositol 3,4,5-triphosphate; PI3K, phosphoinositide 3-kinase.

**curve)**. The polarization of the Rac reporter, however, showed different dynamics (**Fig 3D, green curve)**. Upon initial activation (time = 0), the Rac signal becomes transiently hyperpolarized before stabilizing. During the subsequent recovery and global activation phases of the assay, the Rac signal overshoots and becomes slightly polarized toward the previously unstimulated side of the cell before trending back toward a nonpolarized state. An explanation for this Rac-specific polarity reversal is the presence of one or more local Rac inhibitor(s) accumulating at the prestimulus site, accounting for both the early decline from hyperpolarization and the overshooting behavior observed later.

By applying temporally spaced global pulses in latrunculin-treated cells, we identified a 90-second recovery half-time of the Rac response (**S8 Fig**). Since cells in this latrunculin-treated context do not exhibit global inhibition, this recovery time instead appears to be related to the timescale of recovery from local inhibition (**Fig 3A and 3B**). Within its full signaling context (in the absence of latrunculin), this local inhibition of Rac activity likely represents a Meinhardt-style local inhibitor, preventing front-locking associated with local-positive-feedback, global-negative-feedback systems.

## Local inhibition operates through negative feedback, linking front signaling to input rates

Beyond its potential role in avoiding the locking of cell polarity decisions, we examined how local Rac inhibition could influence a cell's interpretation of guidance cues over time. Depending on the signaling topology of the inhibition, different strategies of cell guidance can be achieved. [44–48]. For example, bacteria chemotax by using integral negative feedback to temporally sense spatial signal gradients while they move through them [49,50]. The internally measured rate of change of the external signal is used to modulate tumbling frequency and accomplish a biased random walk [51]. We wondered whether the local front-based inhibition that we identified might play an analogous role in neutrophil interpretation of guidance cues through local temporal sensing. To investigate this possibility, we took advantage of the titratability and controllability of our system to understand how PIP$_3$ input dynamics regulate the Rac response. If Rac operates through an integral negative feedback circuit, it should respond to the rate of change in PIP$_3$ (analogously to [52,53]).

When we exposed latrunculin-treated cells to a rapid increase in PIP$_3$, there was a transient increase in Rac activation (**Fig 4A, left**). We independently confirmed this behavior using an ELISA-like Rac activation assay (Rac 1,2,3 GLISA assay, Cytoskeleton) (**S9A and S9B Fig**). We took advantage of the precision of our microscopy-based assay to expose cells to a slowly ramped increase in PIP$_3$ rather than a sudden increase, eventually reaching the same steady state. In this case, we observed little to no Rac activation (**Fig 4A, right**). Thus, the Rac response in these conditions appears to depend on the rate of change of PIP$_3$ rather than absolute levels of PIP$_3$.

Rate-sensitive responses are a general feature of negative feedback–based adaptive systems, and the Rac activity response to differing PIP$_3$ input dynamics is consistent with adaptation through negative feedback, as opposed to other circuit topologies like incoherent feedforward [44–48]. To further evaluate whether our system is operating through negative feedback, we tested the effect of Rac inhibition on the dynamics of the response. If the system operates through a Rac-dependent negative feedback loop, then inhibition of Rac's ability to signal to

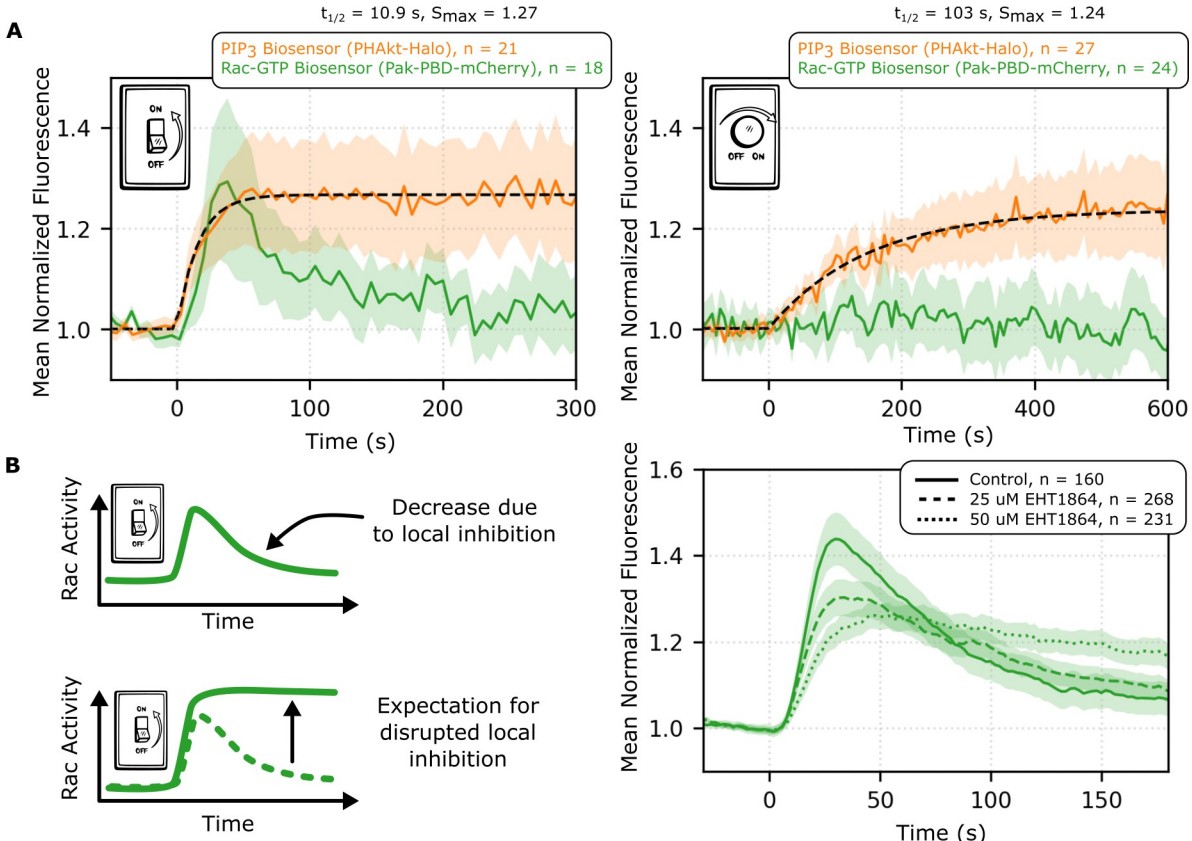

**Fig 4. Local inhibition on Rac signaling enables rate detection that operates through negative feedback. (A)** When latrunculin-treated opto-PI3K cells are exposed to a sudden, step-like increase in blue light (left panel), the PIP$_3$ biosensor rapidly approaches steady state, and the Rac biosensor has a rapid increase followed by a decline. In contrast, when the light levels are increased slowly (right panel), the PIP$_3$ biosensor slowly accumulates to a similar steady state as the step input, but the Rac biosensor response is dramatically attenuated compared to the Rac response seen following a step input of PIP$_3$. Rac dynamics are thus correlated with the rate of change of PIP$_3$ rather than the absolute levels of PIP$_3$ in this context. Mean local responses shown with 95% CI shaded. **(B)** Treating cells with EHT1864, an inhibitor that prevents Rac from signaling to downstream effectors, disrupts the decline in Rac activity that eventually follows a step-like increase in PIP$_3$. This suggests that Rac activity is essential for Rac inhibition—a negative feedback loop. Adaptation via negative feedback enables global rate-sensing in bacterial chemotaxis, and this may represent a similar strategy for interpreting guidance cues in amoeboid chemotaxis. Mean responses shown with 95% CI shaded. The underlying data for this figure can be found in **S1 Data**. PIP$_3$, phosphatidylinositol 3,4,5-triphosphate; PI3K, phosphoinositide 3-kinase.

its effectors should disrupt Rac negative feedback and impair adaptation. We inhibited the ability of Rac to signal to its downstream effectors by treating cells with EHT1864 [54] and observed a dose-dependent shift from transient to sustained Rac activation following an opto-PI3K step input (**Fig 4B**). (See **S9C and S9D Fig** for more details of the dose response.) These data suggest that Rac activity is required for Rac inhibition, consistent with Rac-induced Rac negative feedback.

Though Pak-PBD primarily recognizes active Rac in HL-60 cells [35], the domain also has affinity for Cdc42 in vitro and in cells [55,56]. This could complicate our interpretation of Pak-PBD–based measurements. To test whether our biosensor-based results were Rac specific, we verified the transient Rac activation in response to opto-PI3K stimulation with a Rac-specific GLISA assay (**S9A and S9B Fig**), as mentioned previously. Additionally, we repeated a subset of our experiments in Cdc42-null PLB cells expressing the opto-PI3K system and the Pak-PBD biosensor (PLB cells are a subclone of HL-60 cells, a kind gift from Sean Collins). We found that Cdc42 is not required for the opto-PI3K–induced transient Pak-PBD response

(**S10A Fig**), nor is it required for opto-PI3K–directed guidance (**S10B and S10C Fig**). Though Cdc42 is an important regulator of cell movement under physiological conditions [22,57,58], optogenetic control of PI3K signaling may isolate a Rac-dominant axis of signaling. The principles learned from this particular signaling axis may reveal general principles of signaling organization across other axes as well.

By operating through adaptive negative feedback, a Meinhardt-style local inhibitor in amoeboid cells could act analogously to the negative feedback system used by bacteria to accomplish chemotaxis through rate measurement. This rate dependence is a proposed behavior in the pilot pseudopod model for gradient interpretation, as initially proposed by Gerisch [25]. The pilot pseudopod model proposes that migrating cells can navigate chemoattractant gradients across many orders of magnitude by maintaining (randomly initiated) pseudopodia that experience a temporal increase in signal as they extend up a gradient at the expense of those that experience a temporal decrease in signal when extending down a gradient. A Meinhardt-style local inhibitor, operating in an adaptive negative feedback mechanism, could be used to detect rates-of-change in local signals and, thus, mechanistically enable pilot pseudpod behavior.

## Fronts of polarized cells respond to input signal rates, consistent with a pilot pseudopod model

Our work demonstrates that local negative feedback sensitizes biochemical front signals to temporal changes in signaling inputs, consistent with a pilot pseudopod model of gradient interpretation. We returned to our computer vision–based stimulation assays in the context of moving cells to extend our biochemical signaling observations to cell guidance decisions. We sought to disambiguate the role of temporal changes in input signals from their absolute levels by performing an experiment analogous to our shadow experiment (**Fig 3D**). Since the backs of cells are relatively insensitive to stimulation, we focused on stimulation of the lateral edges of migrating cells.

When we persistently activate cells with local opto-PI3K stimulation at one lateral edge, we can direct them to continuously turn clockwise (for right edge stimulation) or counterclockwise (for left edge stimulation. See **S3 Video**). By switching cells from local to global activation, both edges of the cell attain the same final level of signaling but through different histories. The prestimulated edge will have a smaller increase than the stimulus-naive edge (**Fig 5A**). If cells respond to signaling levels without history dependence, they should respond to the global stimulus as they did previously (**Fig 2E, right**). In contrast, if cells respond to temporal changes in input signals (such as through the local Rac negative feedback loop we have demonstrated), we expect cells to reverse directionality due to the larger increase on their stimulus-naive sides. Our experiments confirm the local temporal interpretation of PIP$_3$ inputs (**Fig 5B and S6 Video**) in a manner analogous to our shadow experiments in latrunculin-treated cells (**Fig 3D and S5 Video**).

We estimated the average angular velocity (in rotations per minute) of cells during each of the three phases of the assay across replicates of this assay (**Fig 5C and S7 Video**). We found statistically significant changes in these velocities (**Fig 5D**). The *p*-values shown indicate the likelihood of observing these distributions under the null hypothesis of an average angular velocity of zero. We verified that the PIP$_3$ biosensor matched the expected temporal profile in **Fig 5A** on the population level (**Fig 5E**). Though the left and right sides of migrating cells reach similar steady states, the rates of change differ between the two sides. Turning behaviors of the cells coincide with the temporal changes in the input signal rather than the absolute levels of the signal. As further validation, we measured initial rates of PIP$_3$ accumulation (as

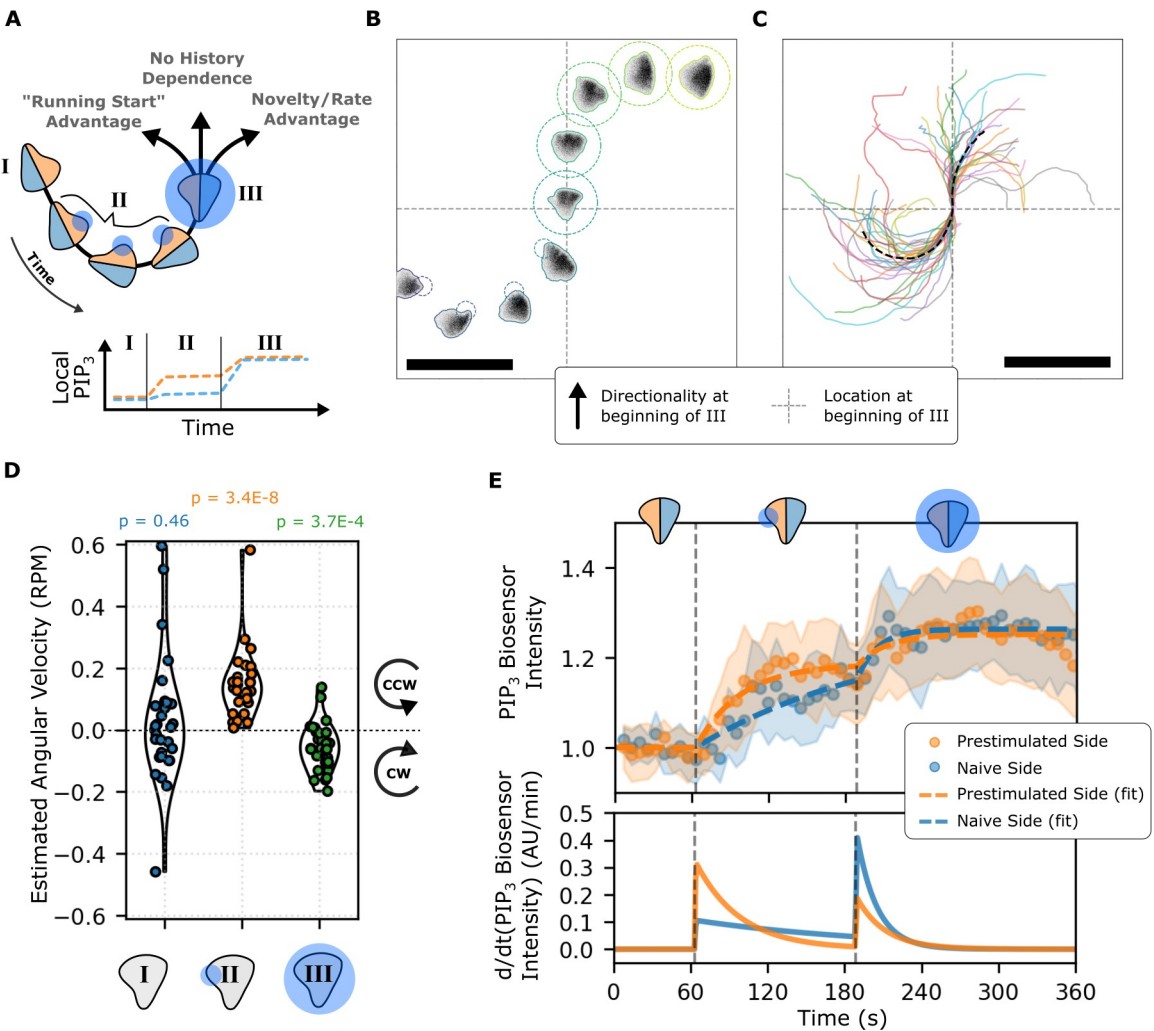

**Fig 5. Cell fronts respond to local temporal changes in input signals, consistent with a modified pilot pseudopod model of cell guidance.** (A) To probe how cells decode the spatial and temporal features of input signals, we designed a test of the pilot pseudopod model. Cells were first tracked in I, then continuously stimulated (via opto-PI3K) at their left edge in II, and finally exposed to global illumination in III. This stimulation regime is expected to cause both sides of the cell to arrive at the same level of $PIP_3$ with different local rates of change, enabling us to determine the relative contributions of spatial and temporal features of input signals to cell orientation. (B) A tracked and stimulated cell responding to the stimulus scheme described in Fig 5A. See S6 Video for an animated version of this subpanel. Scale bar: 50 μm. (C) Tracks of multiple cells ($n$ = 30) responding to the stimulus scheme described in A show a net bias toward reversal following global stimulation. See S7 Video for an animated demonstration of this behavior. Scale bar: 50 μm. (D) Angular velocities were estimated for cells at the beginning of each of the phases depicted in Fig 5A. *P* values shown describe a *t* test for whether the mean of each distribution is zero. Cells transition from counterclockwise motion to clockwise motion following the switch from local stimulation (phase II) to global stimulation (phase III). (E) The $PIP_3$ signals on the left and right sides of migrating cells were calculated and then fit with a time-delayed saturable function. Sample means are shown as dots, and the 95% CI is shown as the shaded area. The fit functions are shown as dashed lines. As expected, both sides approach the same steady state at the end of phase III, but their histories differ. The changes in cell directionality correlate with the local derivative of input signals rather than the absolute levels of the input signal, consistent with a pilot pseudopod model of gradient interpretation. The underlying data for this figure can be found in S1 Data. $PIP_3$, phosphatidylinositol 3,4,5-triphosphate; PI3K, phosphoinositide 3-kinase.

assayed via PHAkt recruitment) on each side for individual cells during the assay. In most cases, the initial rates matched the expected behavior (**S11 Fig**).

By performing an experiment where we subject cells to local temporal changes in $PIP_3$ in the absence of absolute differences in $PIP_3$ across the cell, we demonstrate the sensitivity of

cells to local signaling changes for cell guidance, consistent with a pilot pseudopod model of gradient interpretation.

## Discussion

To navigate toward sites of injury and inflammation, neutrophils must balance the decisiveness required to move in a single direction with the flexibility required to dynamically update this direction. Previous work has shown that cells use local positive feedback [7–9,59] and global negative feedback [5,10,11,42,43] to consolidate protrusive activity in a single direction, but these mechanisms alone do not account for the flexibility of directional orientation [3]. Our current work suggests that Rac-dependent local Rac inhibition plays a key role in neutrophil guidance by enabling this flexibility. Furthermore, our work suggests that this flexibility is closely tied to temporal signal processing through a dynamic negative feedback process. Our data bridge two classic cell migration models (Meinhardt's local inhibition [3] and Gerisch's pilot pseudopod model [25]) and connect them through a modern concept from systems biology (adaptive negative feedback [44–48]).

We leveraged our ability to direct cell migration with optogenetically controlled PI3K (**Fig 1**) to compare the relative responsiveness of different regions of polarized cells (**Fig 2**) and to ask which spatiotemporal features of PIP$_3$ signals inform cell orientation in a two-dimensional environment. Inhibiting both the global negative and local positive feedback loops that organize cell polarity, our optogenetic approach provided direct evidence of local inhibition of Rac activation (as seen in our "shadow" experiments, **Fig 3**). The activity of one or more inhibitors could explain how migrating cells avoid the polarity-locking behavior expected from local positive feedback and global inhibition alone [3]. By controlling the dynamics of PIP$_3$ accumulation in this same global inhibition–inhibited context, we demonstrated that a locally acting Rac-dependent negative feedback loop enables cells to detect the local rate of change of PIP$_3$ (**Fig 4**). Taking into account reduced local responsiveness at the backs of cells (consistent with earlier studies [21,23,26–29]), we leveraged our optogenetic inputs and an imaging-based control system to present cells with inputs that generated local changes in PIP$_3$ in the absence of spatial differences in PIP$_3$ (**Fig 5**). These experiments demonstrate that cells can decode the local rate of stimulus change for cell guidance. Using this model as a framework, in future work, we may be able to build predictive quantitative models of cell behavior and test and fine-tune them on the fly using computer-controlled optogenetics.

Our observations are consistent with the classic pilot pseudopod model, in which the rate of change of stimulus (rather than the absolute concentration of stimulus) is used to influence pseudopod lifetimes for cell guidance during chemotaxis [25]. We extend this model to account for our observations of polarized signal sensitivity (similar to [26]). Rac signals at the center of the front may be saturated or unable to further increase due to the presence of local inhibition. We expect the local negative feedback at the fronts of cells to both keep the front from "locking" and to enable cells to turn toward lateral edges experiencing temporal increases in signal. In much the same way that signals in peripheral vision direct the center of visual attention [60–62], temporal changes in signals at the periphery of the cell front alter its directionality (**Fig 6**).

The sensitivity to temporal input cues does not appear to extend to the backs of cells (**Fig 2**) [21,23,26–29]. Functionally, this insensitivity may help cells to integrate directional choices from the front over time through contractility and actin flow-dependent mechanisms [63,64]. This integration may be particularly important in shallow or changing gradients that may be too "noisy" for cells to accurately interpret instantaneously. For example, in shallow gradients, *Dictyostelium* [65] and zebrafish neutrophils [64] tend to orient themselves through gradient-

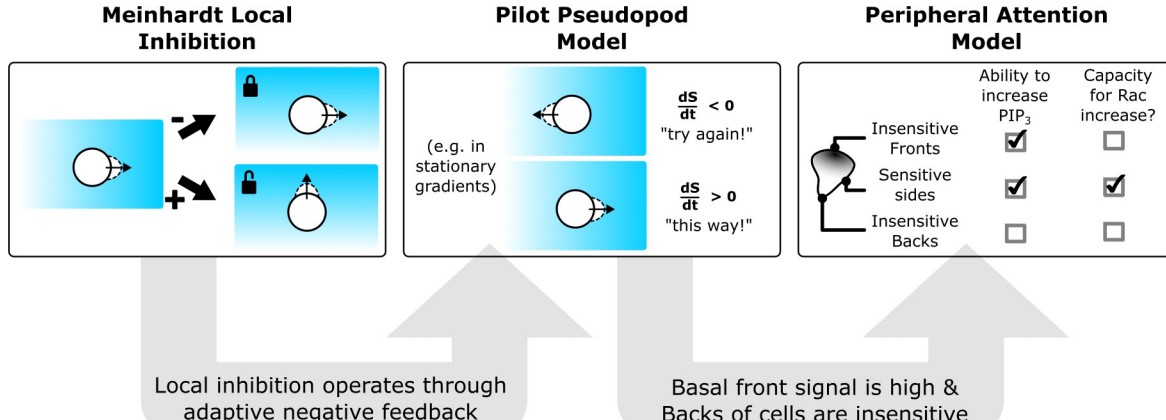

**Fig 6. Cells interpret gradients using a modified pilot pseudopod (peripheral attention) program.** In this work, we have found direct evidence for Meinhardt-style local inhibition operating on the PI3K to Rac signaling axis (first panel). This local Rac negative feedback loop links the rate of change of input signals like PI3K signaling to the magnitude of Rac activation (second panel). This rate sensitivity is an expectation of the pilot pseudopod model. In the more complex case of a migrating cell (third panel), we find that the principles of the pilot pseudopod model apply with two modifications: the first being that the backs of cells are relatively insensitive to stimulation (as previously observed [23,26,27,29]); the second being that the Rac signals at the extreme fronts of cells may already be near-saturation. Taken together, these two modifications make the lateral edges of migrating cells highly sensitive to increases in PIP$_3$, suggesting a role for local inhibition in the ability of cells to turn and refine their direction during chemotaxis.

biased left and right adjustments rather than more dramatic reversals. In these contexts and in our work, cells maintain a rough heading while also retaining the ability to fine-tune directionality through these left–right "peripheral" adjustments.

In this study, we have focused on PIP$_3$ as a model signal because it is sufficient to induce protrusions and to guide migrating HL-60 cells [10,24,30]. Interestingly, the Rac-dependent negative feedback we described may not require PIP$_3$ itself to be the Rac-activating cue. One property of negative feedback–based adaptation, as opposed to an incoherent feedforward-based mechanism, is that it could respond to many activating cues simultaneously. In an incoherent feedforward-based mechanism, a PIP$_3$-dependent mediator would act to shut down Rac signaling. If Rac inhibition operated only in this way, each activator would need an accompanying negative regulatory process. In the negative feedback–based mechanism we described, the activity of the negative regulator is Rac- dependent, so the identity of the activating cue(s) may not matter. This strategy may enable signaling hubs like Rac to act as integrators of temporal information from many simultaneous cues, as cells are likely to experience in situ. This is also consistent with PIP$_3$'s status as a sufficient cue for directing protrusions [10,24,30] that is also not necessary for chemotaxis [65,66].

The hybrid spatiotemporal guidance strategy we have revealed stands in contrast to systems of gradient interpretation that rely on only temporal cues, like those of bacteria. Because of their small size and rapid movement, bacteria use a purely temporal mechanism to migrate toward chemoattractants [51]. Eukaryotic cells are larger and slower and take advantage of spatial signal processing to interpret guidance cues [67]. However, eukaryotic cells do not solely decode gradients spatially but appear to also incorporate temporal information in their guidance [25]. Importantly, the temporal information either could come from cells moving up fixed spatial gradients or could come from dynamically evolving or even self-generated gradients [68–70]. Experiments directly manipulating the spatiotemporal features of soluble chemoattractant gradients have shown that neutrophils reverse direction when they experience a temporal decrease in chemoattractant [71] and that cells require temporal increases in overall signaling during chemotaxis for efficient guidance [72]. Our results suggest that these

behaviors could be explained by the presence of a local Rac inhibitor at the fronts of cells. Future studies examining cell behaviors in stable, dynamic, or self-generated gradients should consider both spatial and temporal features of these gradients as cells navigate them.

## Materials and methods

### Cell culture

HL-60 cells were grown in RPMI 1640 media supplemented with l-glutamine and 25 mM HEPES (Mediatech) and containing 10% (vol/vol) heat-inactivated fetal bovine serum (Gibco). Cultures were maintained at a density of 0.2 to 1.0 million cells/ml at $37°C/5\%$ $CO_2$. HL-60 cells were differentiated by adding 1.5% (vol/vol) DMSO (Sigma-Aldrich) to actively growing cells and incubated for 5 days. HEK293T cells (used to generate lentivirus for transduction of HL-60 cells) were grown in DMEM (Mediatech) containing 10% (vol/vol) heat-inactivated fetal bovine serum and maintained at $37°C/5\%$ $CO_2$.

### Plasmids

Plasmids were assembled using a Golden Gate–based modular cloning toolkit [73,74].

The two key constructs for opto-PI3K system are based on the iLID system [31] and include a mTagBFP2-tagged, membrane-localized, light-sensitive component: AsLov2-SsrA-mTagBFP2-CAAX and an EGFP-tagged translocatable PI3K-binding component: ISH2-EGFP-SspBMicro. Upon 470-nm light exposure, the translocatable component localizes to the plasma membrane, and the ISH2 domain recruits endogenous PI3K to the plasma membrane [8,37].

Biosensors for $PIP_3$ [32–34] and active Rac [35] were also created using the modular cloning kit and were designed to be continuously imaged during experiments without activating the blue light–sensitive optogenetic system. For experiments shown in this paper, $PIP_3$ was detected using a Halo-tagged [75] PH domain from Akt labeled with Janelia Fluor 646 [76], and active Rac was detected using an mCherry-tagged Pak-PBD domain from Pak1.

### Transduction of HL-60 cells

HEK293T cells were seeded into 6-well plates and grown until about 80% confluent. For each well, 1.5 μg pHR vector (containing the appropriate transgene), 0.167 μg vesicular stomatitis virus-G vector, and 1.2 μg cytomegalovirus 8.91 vector were mixed and prepared for transfection using TransIT-293 transfection reagent (Mirus Bio) per the manufacturer's instructions. After transfection, cells were grown for additional 3 days, after which virus-containing supernatants were harvested and concentrated 20-fold using a Lenti-X Concentrator (Takara Bio) per the manufacturer's instructions. Concentrated viruses were frozen and stored at $-80°C$ until needed. For all transductions, the thawed virus was mixed with $3 \times 10^5$ cells in growth media supplemented with polybrene (8 μg/ml) and incubated overnight. Cells expressing desired transgenes were isolated by FACS.

### Preparation of cells for microscopy

Cells expressing Halo fusion proteins were stained with 10 nM JF646 (Janelia) for 10 to 15 minutes at $37°C$ in complete media and then rinsed once with complete media (RPMI with 10% FBS) before placing cells in reduced-serum media for migration assays.

We used an under-agarose preparation [77] with reduced serum (RPMI with 2% serum) to keep cells adjacent to the coverglass for TIRF imaging and optogenetic stimulation. This

preparation involved layering 2% to 2.5% low melting point agarose onto cells after they had been allowed to attach to the glass for 10 to 15 minutes.

## Microscopy hardware

Hardware used for the experiments included an Eclipse Ti inverted microscope equipped with a motorized laser TIRF illumination unit, a Borealis beam-conditioning unit (Andor Technology), a 60× Plan Apochromat TIRF 1.49 NA objective (Nikon), an iXon Ultra electron-multiplying charge-coupled device camera, and a laser merge module (LMM5; Spectral Applied Research) equipped with 405-, 488-, 561-, and 640-nm laser lines. All hardware was controlled using Micro-Manager [36,38] (University of California, San Francisco), and all experiments were performed at 37˚C and 5%$CO_2$.

Activity of opto-PI3K was controlled via a 470-nm (blue) LED (Lightspeed Technologies) that transmitted light through a custom DMD (Andor Technology) at varying intensities by connecting the LEDs to the analog outputs of a digital-to-analogue converter and setting the LED voltages using serial commands via custom Python code. Our microscope is equipped with two stacked dichroic turrets such that samples can be simultaneously illuminated with LEDs using a 488-nm longpass dichroic filter (Chroma Technology) in the upper turret while also placing the appropriate dichroic (Chroma Technology) in the lower turret for TIRF microscopy.

## Automated microscopy and light activation pipeline

Images were collected, and light was spatially patterned using custom tracking and light-patterning code written in python and making use of Pycromanager [40]. The code we used for our experiments and analysis is available on Zenodo (10.5281/zenodo.8217768).

In brief, a program scanned locations in a prepared well while running a segmentation algorithm at each location. If a segmented object was found to be an appropriate size and intensity, the program would then image that object for 30 seconds to test whether it was moving at a speed appropriate for a migrating cell. If so, it moved that object into the center of the field of view and initiated a tracking and stimulus protocol to track the cell over time and deliver programmed stimuli described in this paper.

## Image and data analysis

**Segmentation and background correction.** Collected TIRF images were segmented based on fluorescence intensity from the PHAkt channel. Background corrections were accomplished for each channel by first segmenting the backgrounds of images, then fitting either a plane

$$z = ax + by + c \tag{1}$$

(for 400 px-400 px cropped images) or a two-dimensional quadratic equation

$$z = ax^2 + by^2 + cxy + dx + ey + f \tag{2}$$

(for full size 1,024 px-1,024 px images) to the selected background pixel intensities. After background subtraction, if a cell's average intensity was less than 100 to 200 units, it was excluded from analysis due to unreliable baseline normalization.

**Radial spatiotemporal biosensor quantification.** For quantification of biosensor dynamics around the peripheries of single cells, we first segmented single cells and background-subtracted the images. We then contracted the segmentation mask by 14 pixels (3.0 μm) and

subtracted the remaining center to get a ring-like mask around the peiphery of the cell for each time point. We then collected the coordinates of pixels within this mask, calculated the angle to the centroid for each pixel (subtracting the angle representing initial directionality), and took binned averages around this periphery. This resulted in a 1D vector description of the signal around the edges of the cells for each time point. These vectors were stacked to produce a 2D kymograph-like quantification, and these kymograph-like data were averaged across cells within a particular assay to produce the visualizations shown in **Figs 1E, 1F, S3, and S4**. For **S4B Fig**, we reflected the single cell kymographs if a cell's final position in the x direction was negative and then performed the same averaging procedure with these aligned datasets. The data in **Fig 2B** come from this aligned dataset in **S4B Fig** from the columns at ±90˚.

**Classification of movement toward frontward and rearward stimuli in ±90˚ two-spot assays.** The distance traveled in each stimulation direction in **Fig 2C** was calculated and classified as going toward either the frontward stimulus if the cell moved at least 20 μm in that direction, the rearward stimulus if the cell moved at least 20 μm in that direction, or neither if the cell did not move more than 20 μm in either direction. Cells moved in a stable fashion, similar to **Fig 2A**. The "neither" cells were nonresponders that typically appeared to have lost expression of one or both of the optogenetic components. Single cell traces and classification scheme can be seen in **S5A Fig**. We also checked the robustness of our results by increasing the 20-μm cutoff to 50 μm (**S5B Fig**) and found that this did not significantly change the outcomes of the assay.

**Detection of reversals in ±45˚ and ±90˚ two-spot assays.** To detect reversals in directionality during the two-spot assays shown in **Fig 2D** and **S6 Fig**, we first slightly smoothed the cell trajectories in x and y using a Savitzky–Golay filter (window length of 9 and polynomial order 3) and then calculated the angle and displacement between successive frames. We disregarded angles from time points where the displacement was less than 2 μm/min as these were unreliable measurements. Instead, we assume these values were unchanged from the last confidently measured angle. Finally, we "unwrapped" the values, which account for the periodicity of angular measurements, converting the values into accumulative measurements by adding or subtracting multiples of $2\pi$ where necessary. For each cell, we then classified movement as being aligned with a stimulus if it was within 0.4 radians of the angle of that stimulus (in either direction). If cells spent any time classified as being aligned with one stimulus and the other during the same continuous period of stimulation, they were considered reversers.

**$PIP_3$ and Rac biosensor left/right quantification and polarization metrics.** To calculate the biosensor dynamics on the left and right sides of cells in **Fig 3C**, we segmented cells and split the mask into left and right halves based on whether points were to the left or right of the geometric centroid of the mask. Using these masks, we collected background-subtracted pixel intensities across experiments and normalized them to each cell's baseline fluorescence. We then averaged these together to produce the data shown in **Fig 3C**.

As a metric for calculating the directionality of the Rac biosensor distribution in **Fig 3D**, we calculated the background-subtracted average normalized biosensor dynamics on the left (prestimulated) and right sides similar to the data shown in **Fig 3C** and then took the difference between the left and right sides as a simple polarity metric. We then averaged these single cell dynamics together and calculated confidence intervals.

**Global biosensor quantification.** For quantification of biosensor dynamics in single cells (as in **Figs 4, S8B, S9D, and S10A**), we estimated the centers of cells by blurring the image and using a peak finding algorithm (the peak_local_max function from the feature module of Scikit-image v.0.18.3) [78]. We then used a 20 by 20 pixel (4.3 by 4.3 μm) square centered at those locations as a sampled approximation of whole-cell dynamics. We next took background-subtracted single cell average traces and divided each one by the baseline prestimulus intensity

(we excluded cells with a baseline background-subtracted fluorescence below 100 units to avoid errors from dividing by small numbers). These normalized traces were then averaged together. Corresponding timestamps were extracted from Micromanager metadata.

**Estimation of angular velocity.** To estimate angular velocity of continuously turning cells shown in **Fig 5D**, we processed the angular movement of cells as described above for detection of reversals in two-spot assays. We then fit a line to the angle values as functions of time, recorded the slope values, and converted them into revolutions per minute. For prestimulus velocities (I), the entire 1-minute prestimulus time period was used. For the local stimulus velocities (II), the 2-minute period immediately preceding the switch to global stimulation was used. For the global stimulus velocities (III), the 2-minute period immediately following the switch to global stimulation was used.

**Determination of spatial PHAkt dynamics in migrating cells and estimation of initial rates.** To estimate the PIP$_3$ dynamics on the left and right sides of cells as they were stimulated according to the schematic in **Fig 5A**, we calculated the cell direction and used that as an axis to define the left and right sides of the cells. We calculated the background-corrected mean fluorescence signals on each side over time and averaged the global baseline–corrected dynamics to produce **Fig 5E**. We fit these dynamics to Eq (5) to better estimate the rates of accumulation on each side. We estimated rates for individual cells on their left and right halves by fitting a line to the signal during the first 30 seconds of stimulus response.

**Fitting curves to PHakt dynamics.** To estimate half-times for PIP$_3$ accumulation, we fit a piecewise dynamic model to the population-averaged, background-subtracted, and baseline-normalized PHAkt fluorescence data using scipy's curve fit function. The equation for **Fig 4A** was

$$f(t) = \begin{cases} 1, & \text{if } t < \tau \\ s - (s-1)e^{b(t-\tau)}, & \text{if } t \geq \tau \end{cases} \tag{3}$$

where s is the saturation value, $\tau$ represents an adjustable delay, and b is related to the half-time:

$$t_{1/2} = \frac{-ln(2)}{b}. \tag{4}$$

Similarly, for the curves shown in **Fig 5E**, we used a two-stage saturable function:

$$f(t) = \begin{cases} 1, & \text{if } t < \tau_1 \\ s_1 - (s_1 - 1)e^{b_1(t-\tau_1)}, & \text{if } \tau_1 \geq t \geq \tau_2 \\ s_2 - (s_2 - v)e^{b_2(t-\tau_2)}, & \text{if } t > \tau_2 \end{cases} \tag{5}$$

where

$$v = s_1 - (s_1 - 1)e^{b_1(\tau_2 - \tau_1)}. \tag{6}$$

## Measurement of Rac-GTP by GLISA

For biochemical verification of Rac-GTP dynamics in response to acute optogenetic activation of PIP$_3$, we used a colorimetric Rac 1,2,3-GLISA (Cytoskeleton).

After testing several cell densities to determine the linear range of the assay, we settled on using $2 \times 10^5$ undifferentiated HL-60 cells per sample. So, $2 \times 10^5$ cells in 10 μL DPBS were placed in centrifuge tubes and exposed to a saturating dose of blue LED light for varying amounts of time. Cells were lysed in 120 μL ice cold GL36 buffer, and 100 μL lysate was transferred into ice-cold tubes and immediately snap-frozen in liquid nitrogen. The remaining

20 μL were used to verify that each of the samples had similar protein concentrations (of approximately 0.16 mg/mL).

Once all samples were collected, the 100 μL aliquots were thawed in a room temperature water bath and processed according to the manufacturer's instructions.

## Supporting information

**S1 Fig. Interrogation of cellular decision-making with computer-automated optogenetic control of PI3K. (A)** To better understand how migrating cells make directional decisions, we used an optogenetic strategy for spatial and temporal control over PIP$_3$, a signal that is sufficient to induce protrusion generation. Using TIRF microscopy, live-cell tracking, and computer vision–based automation, we applied various spatiotemporal dynamics of PIP$_3$ generation to migrating cells and observed their signaling and migration responses. We used this strategy to identify the spatial and temporal features of input signals that control cell guidance. **(B)** TIRF microscopy of a 10-μM latrunculin-treated HL-60 cell that was exposed to blue light to activate opto-PI3K, first the left side and then the right side of the cell. **(C)** Schematic of quantification scheme. Background-subtracted pixel intensities within the segmented region (enclosed by red-dashed line) were binned based on their x position and then the average of each bin was collected to produce one vector per time point representing the average left-to-right fluorescence signal. **(D)** Kymograph of the PIP$_3$ biosensor fluorescence signal (PHAkt-Halo (JF646)) of time and the lateral position of pixels (as quantified in S1C). The fluorescence signal closely corresponds to the dynamics of the blue light activation regions (enclosed within the blue outlines). The underlying data for this figure can be found in **S1 Data**.
(TIFF)

**S2 Fig. Characterization of cellular directional responses to automated patterns of optogenetic PI3K stimulation.** In each subpanel, the spatial data have been rotated and translated such that the cell displacement in the first minute of imaging (prestimulus) is toward the top of the figure, and the location of the cell at the zero second mark (moment of stimulation) is at the intersection of the gray, dashed lines. In each panel, Pak-PBD-mCherry (Rac biosensor) TIRF signal is shown in the Single Cell Assay subpanels. Scale bars in all subpanels: 50 μm **(A)** Migrating, opto-PI3K–expressing HL-60 cells were exposed to local blue light exposure at their fronts. The stimulus was centered at the cell edge at 0˚ relative to the displacement of the cell during the first minute of observation. The paths of 45 cells are shown (right). This pattern of stimulation causes hyperpersistent movement in the direction of stimulation. **(B)** Migrating, opto-PI3K–expressing HL-60 cells were exposed to uniform blue light exposure along their bottom surfaces. The paths of 39 cells are shown (right). This pattern tends to cause cells to deviate slightly to the right or left relative to their initial direction (see **Fig 2E**). **(C)** Migrating, opto-PI3K–expressing HL-60 cells were exposed to local blue light exposure at their backs. The paths of 17 cells are shown (right). This pattern of stimulation tended to cause cells to perform "u-turns." **(D)** The angles of displacement for each cell were calculated, and cosine of the difference between these angles and the angle of stimulus were then calculated for each assay type as an indicator of stimulus alignment dynamics. Interestingly, back-stimulated cell alignment dynamics do not appear to match side-stimulated dynamics, even when they are spatially identical (i.e., when the green curve passes 0 on the y-axis). This is likely due to the spatial insensitivity at the cell back and the resulting slow local accumulation of PIP$_3$ in this assay. Mean responses are shown with 95% CI shaded. The underlying data for this figure can be found in **S1 Data**.
(TIFF)

**S3 Fig. Characterization of average spatial biosensor responses to automated patterns of optogenetic PI3K stimulation.** (Related to **2E**) In panels A and B, a polar kymograph shows the average signal of interest around the segmented cell periphery over time during the course of an automated optogenetic stimulation protocol. **(A)** Local opto-PI3K stimulation at the fronts of migrating cells causes a localized increase in PHAkt-Halo localization (PIP$_3$ biosensor) within the area of stimulation. Pak-PBD-mCherry (Rac biosensor) remains hyperpolarized throughout the course of the assay. **(B)** Global opto-PI3K stimulation causes a global increase in PHAkt-Halo (PIP$_3$ biosensor) throughout the cell and a lateral spread in Pak-PBD-mCherry localization. Both signals stay relatively polarized near the 0˚ mark. **(C)** Fold-change in PIP$_3$ and Rac reporters as a function of angle in the front-stimulated cell experiments. These are the means of single-cell ratios where each ratio is the average signal 1 minute poststimulus over the average signal 1 minute prestimulus. The PIP$_3$ signal shows again that we have spatial control over our signaling input. The Rac signal shows evidence of saturation at the center of the fronts of cells (0˚) but shows increases at the edges of the front. Mean responses shown with 95% CI shaded. **(D)** Similar to S3C, this panel shows the fold change in biosensors around the peripheries of cells during the first minute of stimulation. These curves show the mean responses of globally stimulated cells, which show the same saturation phenomenon at the fronts of cells (0˚), while maintaining a capacity for increases at the edges of this front. Mean responses are shown with 95% CI shaded. The underlying data for this figure can be found in **S1 Data**.
(TIFF)

**S4 Fig. Directional and biosensor responses to lateral stimulation. (A)** When opto-PI3K–expressing cells are locally stimulated with blue light at ±90˚, they stably orient toward only one of the two stimuli. Edge kymographs of averaged biosensor signals, as performed previously (**Fig 1**), are difficult to interpret in this assay. Therefore, we aligned cells to the "winning side" by horizontally reflecting cells that migrated left. **(B)** Edge kymographs of aligned cells show that we can locally produce PIP$_3$ and that it is initially similar on both the "winning" and the "losing" sides. The data in **Fig 2B** come from the columns at ±90˚ in the Average PIP$_3$ Response panel. The underlying data for this figure can be found in **S1 Data**.
(TIFF)

**S5 Fig. Migration and classification of frontward-turning cells in angled 180˚-opposed two-spot assays. (A)** Migration paths: For each tested stimulation angle, migrating cells were exposed to two sites of activation simultaneously. These sites were oriented at the indicated angle relative to the initial direction of movement. Those that had migrated 20 μm or more in the direction of the frontward stimulus at the end of the 5-minute assay were classified as moving frontward (blue lines), while those that migrated 20 μm or more in the direction of a rearward stimulus were classified as moving rearward (red lines). All other cells were classified as nonresponders (black lines). Shaded regions indicate the classification regions. Scale bars, 50 μm. **(B)** Robustness analysis: We chose to use 20 μm as our cutoff in **Fig 2C**, but the results hold for other distance cutoffs as well. For example, this shows the proportions as in **Fig 2C**, but with a 50-μm threshold for the classification rather than 20 μm. Scale bar, 50 μm. Proportions shown with 95% CI error bars. The underlying data for this figure can be found in **S1 Data**.
(TIFF)

**S6 Fig. Classification of reversing and nonreversing cells in two-spot assays. (A)** Data processing pipeline for reversal classification. Full details are included in the methods section. **(B)**

Migrating cells were stimulated with two local opto-PI3K stimuli located at either ±90° or ±45° relative to the initial direction of migration. Few cells stimulated with the ±90° pattern exhibit reversals. About half of cells stimulated with ±45° stimulus pattern exhibit reversals during the 5-minute assay. Scale bars, 50 μm. The underlying data for this figure can be found in **S1 Data**.
(TIFF)

**S7 Fig. Local sites of PIP$_3$ activity fail to compete with one another in latrunculin-treated cells.** By comparing the Rac biosensor dynamics on the left and right sides of cells, we tested for the winner-take-all behavior expected in the presence of a global inhibitor. Each point is the relative Rac enrichment (compared to baseline) at the end of a 5-minute two-spot opto-PI3K activation assay. **(A)** In control cells stimulated with opto-PI3K at both their left and right edges (relative to their initial directionality), one site becomes Rac-high and one becomes Rac-low, leading to anticorrelated Rac distributions between the two sides. **(B)** In latrunculin-treated cells stimulated on both their right and left edges with opto-PI3K, this winner-take-all system is not functional, and the two sides now show a positive correlation of Rac activity levels. The underlying data for this figure can be found in **S1 Data**.
(TIFF)

**S8 Fig. Timescale of reversibility of Rac inhibition following opto-PI3K stimulation. (A)** Experimental schematic: Latrunculin-treated (10 μM) cells were exposed to two pulses of opto-PI3K activation separated by a variable amount of recovery time. We then calculated the ratio of the peaks heights for the first and second Rac responses. **(B)** Two example curves show the response of the PIP$_3$ and Rac biosensors to two pulses of blue light activation, with recovery times of 60 seconds (top) and 240 seconds (bottom). Sample averages ± 95% CI are shown. **(C)** Degree of recovery as a function of recovery time. The Rac response has a recovery half-time of 89 seconds following opto-PI3K stimulation. The underlying data for this figure can be found in **S1 Data**.
(TIFF)

**S9 Fig. Confirmation of live-cell Rac biosensor dynamics with a Rac GLISA. (A)** Schematic of the ELISA-like mechanism used in the assay. Plate-bound Pak-PBD binds only the active form of cellular Rac (Rac-GTP). This bound active Rac can then be detected and quantified through a colorimetric assay using an HRP-conjugated Rac antibody. **(B)** GLISA-based measurements of relative Rac-GTP in blue-light-exposed opto-PI3K cells closely match the dynamics observed using the live-cell Rac biosensor, Pak-PBD-mCherry (compare with **Fig 4A**, left). Individual replicates are shown in blue, orange, and green. Sample averages ± SD are shown in black. **(C)** Dose–response showing proportion of active Rac in unstimulated opto-PI3K cells (as measured by GLISA) as a function of dose of EHT1864. Individual replicates are shown in blue, orange, and green. Means ± SD are shown in black. **(D)** Dose response of Pak-PBD dynamics in EHT1864-treated cells. Pak-PBD measurements were made via TIRF microscopy and normalized to baselines that were scaled by GLISA results from S9C. Mean responses are shown with 95% CI shaded. The underlying data for this figure can be found in **S1 Data**.
(TIFF)

**S10 Fig. Cdc42 is not required for Pak-PBD polarization or for cellular responses to opto-PI3K stimulation. (A)** Cdc42-null PLB cells exhibit adaptive recruitment of the Rac biosensor PBD following a step input of opto-PI3K, as in wild-type HL-60 cells (compare with **Fig 4A**, left). Opto-PI3K–expressing Cdc42-null cells were stimulated with blue light for 180 seconds, during which time they experienced a decline in the mean Pak-PBD TIRF localization signal.

Mean response is shown with 95% CI shaded. **(B)** Cdc42-null cells maintain their ability to migrate and polarize Pak-PBD. The Cdc42-null cells have broad and relatively unstable fronts that appear to oscillate in width and frequently split. The *p*-value refers to the probability that the average final location of cells is unbiased (i.e., is equal to zero). In the absence of opto-PI3K stimulation, there is no bias. Scale bar for microscopy insets (center): 20 μm. Scale bar for migration paths (right): 50 μm. **(C)** In cells that maintain polarization during the course of optogenetic stimulation, local opto-PI3K stimulation retains its ability to steer Cdc42-null HL-60 cells. Cells were stimulated at 1:00 on their left side. The *p*-value refers to the probability that the average final location of cells is unbiased (i.e., is equal to zero). In the presence of a left-ward opto-PI3K stimulus, there is a significant bias in cell directionality. Area of optogenetic activation shown with blue outline. Scale bar for microscopy insets (center): 20 μm. Scale bar for migration paths (right): 50 μm. The underlying data for this figure can be found in **S1 Data**.
(TIFF)

**S11 Fig. Local PIP$_3$ dynamics in single migrating cells during reversal assay.** Differences in the estimated initial rates of PIP$_3$ reporter (PHAkt-Halo) accumulation on the left and right sides of cells during the reversal assay (**Fig 5**). As expected, the majority of cells show similar PIP$_3$ dynamics on their left and right sides prestimulation (phase I). However, cells show higher rates of PIP$_3$ increase on the left side during the local stimulation phase of the assay (phase II) and higher rates of PIP$_3$ increase on their right sides during the subsequent global stimulation phase of the assay (phase III). The underlying data for this figure can be found in **S1 Data**.
(TIFF)

**S1 Video. PIP$_3$ and Rac biosensors are locally recruited in response to local blue light stimulation.** A single latrunculin-treated (10 μM) cell is shown using TIRF microscopy. The outline of the cell is shown in red, and the outlines of the blue light–stimulated regions are shown in blue. This video corresponds with the data shown in **S1 Fig**.
(MP4)

**S2 Video. Migrating cells respond to single-spot assays by altering their directionality.** Migrating cells were stimulated with spatially patterned blue light (boundary shown with dashed lines). The location of the stimulus was determined automatically using a computer vision pipeline that is available on Zenodo (doi: 10.5281/zenodo.8217768). The live-cell fluorescent Rac biosensor (Pak-PBD-mCherry) is shown. Scale bars are 50 μm. This video corresponds with data shown in **Fig 1D**.
(MP4)

**S3 Video. Computer-controlled local activation can continuously turn migrating cells.** Rather than maintaining an orientation relative to the lab frame of reference (as shown in **S2 Video**), the cell frame of reference was used to continuously target one edge of a migrating cell relative to its direction of movement. Four individually performed assays were extracted and computationally overlayed for this video. The live-cell fluorescent Rac biosensor (Pak-PBD-mCherry) is shown. X and Y axes units are in μm.
(MP4)

**S4 Video. Migrating cells respond to two-spot assays differently depending on the positioning of the spots.** Using the computer-controlled tagreting system, we stimulated cells with two spots of opto-PI3K activation simultaneously. 180˚-opposed spots create a winner-take-all scenario, while 90˚-opposed, frontward-oriented spots cause cells to switch between the two

directions. The live-cell fluorescent Rac biosensor (Pak-PBD-mCherry) is shown. Scale bars are 50 μm. This video corresponds with data shown in **Fig 2D**.
(MP4)

**S5 Video. A local-to-global stimulation assay shows direct evidence of local inhibition of Rac.** Latrunculin-treated opto-PI3K cells (10 μM) were stimulated first locally and then globally with blue light. The prestimulus site has a weaker response during the global stimulation in comparison to the stimulus-naive side. The live-cell fluorescent Rac biosensor (Pak-PBD-mCherry) is shown. This video corresponds with data shown in **Fig 3D**.
(MP4)

**S6 Video. Migrating cell responding to local temporal changes in input signal.** A single migrating cell was continuously stimulated at one edge with opto-PI3K as shown in **S3 Video** and then exposed to global opto-PI3K stimulation in a manner analogous to **S5 Video**. Upon exposure to the global stimulus, the cell turned toward the previously unstimulated side. The live-cell fluorescent Rac biosensor (Pak-PBD-mCherry) is shown. Scale bar is 50 μm. This video corresponds with data shown in **Fig 5B**.
(MP4)

**S7 Video. Multiple migrating cells responding to local temporal changes in PIP$_3$ input signal.** This video shows multiple cells behaving similarly to the one shown in **S6 Video**. The live-cell fluorescent Rac biosensor (Pak-PBD-mCherry) is shown. X and Y axes units are in μm. This video corresponds with data shown in **Fig 5C**.
(MP4)

**S1 Data. Excel file with values used to make plots in all figures.** Exact numerical values generated from the analysis pipelines hosted on Zenodo (doi: 10.5281/zenodo.8217762) using the raw microscopy data hosted on Zenodo (doi: 10.5281/zenodo.8208724). Numerical data are listed in individual spreadsheets for Figs 1C–1F, 2A–2D, 2E (top), 2E (bottom), 3C, 3D, 4A, 4B, 5C, 5E (top, raw averages), 5E (top, fit values), 5E (bottom), S1D, S2A–S2C, S2D, S3A, S3B, S3C, S3D, S4A, S4B, S5B, S6B, S7, S8B (top), S8B (bottom), S8C, S9B, S9C, S9D, S10A, S10B, S10C, and S11. Data are typically presented in long-form format. Error bar "width" in these spreadsheets refers to the value symmetrically added to or subtracted from the mean values to obtain the error bars shown in the figures.
(XLSX)

## Acknowledgments

We thank Anne Pipathsouk and Brian Graziano for helpful discussions; Kirstin Meyer, Ben Winer, and Tamas Nagy for a critical reading of the manuscript; and all members of the Weiner lab for their support. The JF646 HaloTag dye was kindly provided by Dr. Luke Lavis. The Cdc42-null PLB cells were a kind gift from Dr. Sean R. Collins.

## Author Contributions

**Conceptualization:** Jason P. Town, Orion D. Weiner.

**Data curation:** Jason P. Town.

**Formal analysis:** Jason P. Town.

**Funding acquisition:** Jason P. Town, Orion D. Weiner.

**Investigation:** Jason P. Town.

**Methodology:** Jason P. Town.

**Project administration:** Orion D. Weiner.

**Software:** Jason P. Town.

**Supervision:** Orion D. Weiner.

**Visualization:** Jason P. Town.

**Writing – original draft:** Jason P. Town, Orion D. Weiner.

**Writing – review & editing:** Jason P. Town, Orion D. Weiner.

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
