## [Editor Report · Decision Letter 0]

6 Feb 2023

Dear Dr Weiner, 

Thank you for submitting your manuscript entitled "Rac negative feedback links local PIP3 rate-of-change to dynamic control of neutrophil guidance" for consideration as a Research Article by PLOS Biology. Please accept my apologies for the delay in getting back to you as we consulted with an academic editor about your submission. 

Your manuscript has now been evaluated by the PLOS Biology editorial staff, as well as by an academic editor with relevant expertise, and I am writing to let you know that we would like to send your submission out for external peer review.

Once your full submission is complete, your paper will undergo a series of checks in preparation for peer review. After your manuscript has passed the checks it will be sent out for review. To provide the metadata for your submission, please Login to Editorial Manager (https://www.editorialmanager.com/pbiology) within two working days, i.e. by Feb 08 2023 11:59PM.

Kind regards,

Richard

Richard Hodge, PhD

Associate Editor, PLOS Biology

rhodge@plos.org

PLOS

---

## [Decision Letter · Decision Letter 1]

21 Mar 2023

Dear Dr Weiner,

Thank you for your patience while your manuscript "Rac negative feedback links local PIP3 rate-of-change to dynamic control of neutrophil guidance" was peer-reviewed at PLOS Biology. Please accept my apologies for the delays that you have experienced during the peer review process. It has now been evaluated by the PLOS Biology editors, an Academic Editor with relevant expertise, and by four independent reviewers. 

In light of the reviews, which you will find at the end of this email, we would like to invite you to revise the work to thoroughly address the reviewers' reports.

As you will see below, the reviewers are generally positive about your study and think it is interesting and well done. Reviewer #2 notes that the manuscript does not consider the effects of Cdc42 in the model and suggests using Rac/Cdc42-specific FRET probes to differentiate their contributions. In addition, Reviewer #4 raises concerns that the study lacks a deeper underlying mechanism that drives the negative feedback and lists several research directions that could provide these insights. In addition, the reviewers raised overlapping concerns with the presentation of the manuscript and suggest ways in which the data and diagrams used could be more easily followed.

After discussions with the academic editor, we will not make the FRET assays or experiments that provide additional mechanistic insight essential for a revision, but we do ask that you focus on the experimental revisions suggested by Reviewer #1 and strengthen the clarity of the data and its presentation.

Given the extent of revision needed, we cannot make a decision about publication until we have seen the revised manuscript and your response to the reviewers' comments. Your revised manuscript is likely to be sent for further evaluation by all or a subset of the reviewers.

**IMPORTANT - SUBMITTING YOUR REVISION**

*Re-submission Checklist*

*Published Peer Review*

*PLOS Data Policy*

*Blot and Gel Data Policy*

Sincerely,

Richard

Richard Hodge, PhD

Associate Editor, PLOS Biology

rhodge@plos.org

REVIEWS:

Reviewer #1: In this manuscript, Town and Weiner have probed into the decision-making strategy in migrating neutrophil-like HL-60 cells in response to multiple chemoattractant cues. Being the body's first line of defense, neutrophils need to polarize their cytoskeletal regulators toward chemical signals arising from injury and infection. Previous studies have shown that such polarization is brought about by local positive feedback that amplifies the cell front and global negative feedback which allows fronts to compete for dominance and enables the cell to persistently move in a single direction. 

Although this model provides important insight on how cells establish polarity initially, it could not explain how cells can reorient or correct polarity to changing external cues. In this manuscript, the authors have tried to understand how migrating cells balance decisiveness with flexibility in respect to responding to chemoattractant cues. To mimic cellular responses to multiple cues at the same time, authors have employed two-spot optical competition assays using opto-PI3K to acutely increase PIP3 levels at different parts of the cell membrane simultaneously. Their results show that migratory responses varied spatially in polarized cells; in the presence of 180 degrees-opposed stimuli at the lateral edges of the existing front, cells migrated stably in either direction. Thus, they concluded that the front was more responsive to PI3K activation than the back. Once activation spots were moved to 45 degrees from the polarity axis to ensure stimulation would be continuously near the cell fronts, a high percentage of cells showed some degree of reversal between the two orientations. This orientation switching suggested that a local inhibitor at the front could be destabilizing protrusions, as predicted earlier by Meinhardt. The authors then moved ahead to isolate the effects of local inhibition by pharmacologically blocking activation of mechanically-gated global inhibition. Their results showed that local PI3K activation results in recruitment of a reversible and locally acting Rac inhibitor. Finally, the authors suggest that Rac activity is needed for its inhibition which is consistent with Rac-dependent negative feedback.

Overall, this study is technically impressive, however, there are significant gaps that need to be addressed before their main conclusions can be fully supported. In addition, the authors must do a much better job distinguishing the novelty of their findings, given that previous studies (mentioned in the text) have explored the existence of Meinhardt-style inhibitors in cell motility. 

Major concerns:

1. The authors need to show that there is complete absence of global inhibition in presence of latrunculin. A major advancement that the authors claim in their manuscript is that they were able to delineate local inhibition by eliminating global inhibitory effects in migrating cells.

2. The authors claim that local inhibitor works solely on Rac. However, there was no evidence that molecules upstream of Rac, such as PI3K, were inhibited also. In Fig 3C, it is possible that PIP3 is being locally degraded at the first stimulation site.

3. There is very little primary microscopy data in the main figures. This proved to be a hindrance in grasping the main findings of the manuscript. For example, Figure 1 contains many illustrations which could be moved to a supplement to provide room for more actual data. Additionally, the microscopy images in Fig 2C and 5B are far too small to evaluate effectively.

4. Experiments in Fig. 2D and G rely on the technical ability to generate PIP3 at two spots on the membrane separately and simultaneously at similar levels. The authors need to show PIP3 biosensor localization in such an experiment to demonstrate technical feasibility,

Minor Concerns:

1. The flow of the figures and the text is occasionally confusing. For example, the text jumps from Fig 2D to 2G before mentioning the figures in between. Additionally, figure 1 is not referenced in the main text.

2. There are a few places in the text (like line 64) with extra spaces and/or punctuation. Additionally proofreading would improve the clarity of the manuscript.

Reviewer #2: This manuscript characterises spatiotemporal negative feedback mechanisms between PI3K and Rac that control polarisation in DMSO-differentiated HL-60 cells, and presents a new model of this mechanism. Cell biology experiments are presented to support the conclusions. The question being addressed is one that has been discussed, experimented upon and modelled for years by a number of research groups. The new knowledge created might therefore be considered incremental in nature.

A major issue with this paper lies in the fact that Cdc42 is an important regulator of polarisation in mammalian cells, whereas Rac regulates actin polymerization resulting in formation of a protrusion, often called pseudopod in migrating single cells. In neutrophils, as elsewhere, Cdc42 is crucial for polarisation (e.g. Yang et al 2016, Nat Cell Biology 18, 191; Szczur et al 2009, Blood 114, 4527), although it was not considered here. Unfortunately however the 'biosensor' used in the present manuscript to detect active Rac was Pak-PBD, which is not actually specific for Rac-GTP. Rather it detects both GTP-loaded Rac and Cdc42 and may therefore not differentiate between signals created by these two important regulators. Superior and specific FRET probes have long been described and used (e.g. by Yang et al 2016; Johnson et al 2014 Cell Rep 6, 1153) identifying key roles for both Rac and Cdc42 in single cell chemotaxis. To prove their point, the authors should use such a Rac-specific probe. To demonstrate that PI3K and Rac are the only regulators required for neutrophil polarisation as per the model presented here, their work ought to be complemented with a Cdc42 specific probe; if Cdc42 is indeed dispensable, polarisation should also occur in the absence of Cdc42 activation; this could easily be tested experimentally too.

Moreover, since neutrophils (and likely differentiated HL60 cells) express both Rac1 and Rac2 which have overlapping functions in chemotaxis (e.g. Sun et al 2004 Blood 104, 3758), performing actual pull downs with Pak-RBD and probing pull-downs with antibodies specific for Rac1, Rac2 and Cdc42 would be more informative than the G-LISA presented. If G-LISAs are used, a loading control should be shown to ensure that input of the small GTPase in question into all conditions was equal. 

To avoid confusion, the term 'polarisation' should be replaced by alternatives such as pseudopod or protrusion.

Furthermore, HL-60 cells should not be referred to as 'neutrophils' throughout title, abstract etc. because again this is incorrect and in fact confusing for those who do not know better. Neutrophils are terminally differentiated, primary, short-lived innate immune cells that cannot be cultured nor virally transduced. HL-60 cells are not neutrophils; they would better be described as a model system for amoeboid cell chemotaxis. 

Minor points. 

1. Much effort appears to have gone into drawing pretty regulatory diagrams (e.g. Fig1) however to a general biologist they are not very clear. The paper would likely reach a wider readership if the network and regulatory diagrams provided were more intuitive to general biologists.

2. The methods and figure legends provided are a little scarce. Clearer labelling of the regulatory diagrams in the legends might make them easier to understand. Also information should be provided to indicate on how many separate instances experiments were performed. What was the percentage of cells that moved in these experiments? How often were images acquired during cell migration experiments? 

3. Line 366-7, cells were isolated by FACS _sorting_

4. Line 490, 2 x 10^5 cells?

5. EHT1864 has been used at 5um in cell based experiments (Shutes et al 2007 JBC 282, 35666), suggesting that 25/50uM doses are not in fact 'mild inhibitions'. An activity assay would identify whether inhibition at the doses used was indeed incomplete. If these doses achieved complete inhibition, an alternative explanation ought to be provided.

Reviewer #3: This excellent study tackles the problem of spatio-temporal signaling in neutrophil chemotaxis. The authors use an optogenetic system to spatio-temporally manipulate PIP3 input signaling, and record PIP3 and Rac signals using biosensors. The big novelty is that they use a computer vision pipeline to do real time image analysis, and program the microscope to provide specific light dependent PIP3 inputs depending on specific signaling states that are a analyzed on the fly. This level of automation allows them to tackle very sophisticated experiments that are very well thought, and that originate on the knowledge that the Weiner lab has accumulated during years of research.

I think that it is fair to say that the technological approach is a tour de force, and that the questions that have been tackled have been extremely well formulated ! This reviewer congratulates the authors for their excellent work.

Now I must admit that I also have some reservations about the results have been spelled out. First, I am not an expert on immune cell chemotaxis, and it was very difficult to navigate the deluge of different chemotaxis models that are the basis for this work that then are crystallized out in the network circuitry in figure 1B ! Given this lack of knowledge, it was difficult for me to completely understand how the findings were able to revise the models. I guess the authors are so familiar with the different models that it is difficult for them to spell out the essence to a lay audience (which I understand is difficult). Further, I had the feeling that the figures are very conceptual, and often very succinctly described in the legend. I often had to search the right information about a specific experiment in the text and in the figure legends, which made the manuscript difficult to read. By example the model in figure 1B which depicts a complex feedback circuitry was difficult to understand for me. I missed the AND gate in the text. Maybe the authors could refer to this network in the introduction when they are spelling out the different models, and provide information about each feedback. Also, the network circuitries on the right of figure 2B are not well described in the legends. Another prime example of this is figure 4B. On the network scheme on the left there is a reference to "expected effect of latrunculin". There was no reference to latrunculin in the text. Further, the figure legend relevant to Figure 4B was very conceptual, and did not describe the figure panels ! This applies also to many other figures. I had a bit the feeling that the authors put their conceptual powerpoint slides as figures, but that details about experiments were missing. This made the reading of this manuscript difficult ! I would therefore strongly advise the authors to edit the manuscript to better explain the concepts and the results, so that the paper is smoother to read ! I understand that this will make the paper longer to read.

Finally, I think that a weakness of the paper is that it lacks a strong modelling component (similar to the work of Iglesias/Devreotes) that can provide better intuition about the results. I think the authors convincingly demonstrate the existence of a negative feedback at the front, but at the same time I feel that the non-linear behavior of such large signaling feedback circuitries can produce signaling behaviors that might escape the intuition of the authors. The modelling part is indeed beyond the scope of this excellent paper. But I would suggest to the authors to mention in the discussion section that this is now the best interpretation of their results, but that mathematical models will allow to refine this. The sophistication and experimental throughput provided by their automated microscopy approach have the power to realize the promise of mathematical modelling.

I congratulate the authors on their excellent work. I advise them to re-edit the manuscript to make it easier to read, and then I will be ready accept the manuscript !

Reviewer #4: In response to the cues such as injury or infection, neutrophils polarize their cytoskeletons to migrate directionally. Neutrophils achieve this polarization by amplifying leading edge signals through local positive feedback at the front and global negative feedback. However, these two-component models fail to account for the ability of cells to reorient in response to changing cues. Here, the authors use optogenetics to spatially and temporally control leading edge signaling, a computer-vision-based feedback system, a Rac activity sensor, and modeling to ask how cells polarize and reorient to new cues. 

Using an optogenetic PI3K, the authors assess how neutrophils respond to local changes in PI3K activity. They find that cell fronts are more sensitive to than cell backs, and that competing signals near the fronts of cells can induce direction switching. Using latrunculin A treatment, they demonstrate that Rac is activated locally. But after recovery followed by global activation, Rac activity is lowest where it was previously highest. This leads to the proposal that there is local, temporally delayed, negative feedback.

By inducing fast and slow switches in PI3K activity, they show that Rac responds to the rate of PI3K activation. Further, they show that a prior local stimulation results in a change in direction upon global stimulation. Thus, the data suggest that cells move in the direction of the unstimulated side because this side has a larger temporal increase in stimulus compared to the stimulated side.

The authors thus propose that neutrophils employ local negative feedback onto Rac to prevent "locking" onto a single direction and direct '"peripheral attention" to balance "decisiveness" and "flexibility" during chemotaxis. The study design is creative and the experimental approaches are sophisticated. The results presented are mostly clear, informative, and relevant for readers of PLOS Biology. However, the presentation could be clearer and some attempt to uncover a mechanism driving this local negative feedback would be welcome, as described below. 

Major: 

What kinds of mechanisms do the authors think could be operating? Could it simply be that Rac turns itself off over time by hydrolyzing GTP to GDP? Or does the negative feedback require that PI3K, PIP3, RacGTP, or something else recruit a GAP to accelerate Rac inactivation (as implicated in the text? "Together these data suggest that local PIP3 generation leads to recruitment of a reversible, locally acting, moderately persistent Rac inhibitor." Could the effect be due to local depletion of an activator or a refractory period for an activator, internalization of Rac itself, or multiple mechanisms? Mutations that affect the Rac GTPase activity or its interaction with GAPs could address some of these possibilities. Even negative results might be interesting if they rule particular possibilities out. If this point is beyond the scope of this study, the authors should not refer to a discrete inhibitor because the nature of the inhibition is completely unknown. Some discussion of possible mechanisms would be helpful as well.

The main experiment showing a reduction in Rac activity at the initial site of activation upon global stimulation is with Latrunculin A treatment. They state that global tension mediated negative feedback might be difficult to disentangle from a local negative inhibitor. However, it also seems important to assess the contribution of the local inhibition in the context of the more dominant global negative feedback in a normally migrating cell. Can the authors look at the Rac activity reporter as they do with the PIP3 sensor in the assay used in Figure 5? What happens to Rac activity in this context? 

The biosensor spatiotemporal dynamics appear to be different depending on the whether the PI3K stimulus was provided at the front, side or back. Have the authors quantified this distribution and whether this affects cell polarization? For example: When the PI3K stimulus was provided at the back of the cell (Figure 2C), the biosensor polarization at the front was not disrupted over several minutes whereas when the stimulus was provided on the side, the turning appeared rapid.]

It would be helpful to show the actin dynamics in LatA treatment and the shadow experiments. 

In Figure 3C, what would the Rac biosensor look like with a two-spot assay stimulation at the lateral edges after recovery? Can the authors use the model to predict the outcome in such an experimental scenario and then test the prediction? 

Have the authors noticed differences in protrusion lifetimes in the shadow experiments in Figure 3C and 5B? 

Minor:

Figure 1 is not cited in the text and is purely conceptual. While helpful, could it be condensed and perhaps merged with a figure that contains data?

Line 103, pg4: The authors go back and forth between referring to Fig2G and 2D before E and F. The figures should be arranged and discussed in order. 

Figure 3A needs to be labeled- the reader cannot interpret this with figure or figure legend 

Panels 3D and 3E are not cited in the text. 

Figure 5E - were these measurements performed on multiple experimental replicates? This should be shown to show the variability between experiments. 

Statistical testing and P values should to be added to some figures (specifically noticed Figures 2D, and S5) 

Is there a way to test if the inhibition at one site occurs during reorientation/direction switching of neutrophils without optogenetics? 

Can the authors think of a way to shorten the title to make it clearer? For example "Rac negative feedback" is not very clear. Negative feedback from what to what?

---

## [Editor Report · Decision Letter 2]

28 Jul 2023

Dear Dr Weiner,

Thank you for your patience while we considered your revised manuscript "Local negative feedback underlies a pilot pseudopod-like program for amoeboid guidance" for publication as a Research Article at PLOS Biology. This revised version of your manuscript has been evaluated by the PLOS Biology editors and the Academic Editor.

Based on our Academic Editor's assessment of your revision, I am pleased to say that we are likely to accept this manuscript for publication, provided you satisfactorily address the following data and other policy-related requests that I have provided below (A-E):

(A) We would like to suggest the following modification to the title: 

“"Local negative feedback of Rac1 activity at the leading edge underlies a pseudopod-like program for amoeboid cell guidance”

(B) Thank you very much for already providing the underlying numerical data for the figures presented in the manuscript. However, we note that Fig S9A appears to be mislabeled in the data file, as this refers to a cartoon diagram in the manuscript? 

(C) Per journal policy, as the code that you have generated is important to support the conclusions of your manuscript, we require that you make it available without restrictions upon publication. Please ensure that the code is sufficiently well documented and reusable, and that your Data Statement in the Editorial Manager submission system accurately describes where your code can be found. If depositing code on Github, we ask that you please attach this to Zenodo to ensure long term maintenance and that the deposition is assigned a DOI.

(D) Please also ensure that each of the relevant figure legends in your manuscript include information on *WHERE THE UNDERLYING DATA CAN BE FOUND*, and ensure your supplemental data file/s has a legend.

(E) Please ensure that your Data Statement in the submission system accurately describes where your data can be found and is in final format, as it will be published as written there. As noted before, this includes the deposition for the code. 

We expect to receive your revised manuscript within two weeks. 

*Published Peer Review History*

*Press*

Kind regards,

Richard

Richard Hodge, PhD

rhodge@plos.org

PLOS

---

## [Editor Report · Decision Letter 3]

21 Aug 2023

Dear Dr Weiner,

On behalf of my colleagues and the Academic Editor, Laura Machesky, I am pleased to say that we can accept your manuscript for publication, provided you address any remaining formatting and reporting issues. These will be detailed in an email you should receive within 2-3 business days from our colleagues in the journal operations team; no action is required from you until then. Please note that we will not be able to formally accept your manuscript and schedule it for publication until you have completed any requested changes.

PRESS

Best wishes, 

Richard

Richard Hodge, PhD

rhodge@plos.org

PLOS
